# Integrating image and gene-data with a semi-supervised attention model for prediction of KRAS gene mutation status in non-small cell lung cancer

Yuting Xue[1], Dongxu Zhang[1], Liye Jia[1☯], Wanting Yang[1☯], Juanjuan Zhao[2,3]*, Yan Qiang[1], Long Wang[3], Ying Qiao[4], Huajie Yue[4]

**1** College of Information and Computer, Taiyuan University of Technology, Taiyuan, Shanxi, China, **2** School of Software, Taiyuan University of Technology, Taiyuan, Shanxi, China, **3** College of Information, Jinzhong College of Information, Taiyuan, Shanxi, China, **4** First Hospital of Shanxi Medical University, Taiyuan, Shanxi, China

☯ These authors contributed equally to this work.
* zhaojuanjuan@tyut.edu.cn

**Data Availability Statement:** The data are available from the website https://wiki.cancerimagingarchive.net/display/Public/NSCLC+Radiogenomics. The code for S2MMAM is

## Abstract

KRAS is a pathogenic gene frequently implicated in non-small cell lung cancer (NSCLC). However, biopsy as a diagnostic method has practical limitations. Therefore, it is important to accurately determine the mutation status of the KRAS gene non-invasively by combining NSCLC CT images and genetic data for early diagnosis and subsequent targeted therapy of patients. This paper proposes a Semi-supervised Multimodal Multiscale Attention Model (S²MMAM). S²MMAM comprises a Supervised Multilevel Fusion Segmentation Network (SMF-SN) and a Semi-supervised Multimodal Fusion Classification Network (S²MF-CN). S²MMAM facilitates the execution of the classification task by transferring the useful information captured in SMF-SN to the S²MF-CN to improve the model prediction accuracy. In SMF-SN, we propose a Triple Attention-guided Feature Aggregation module for obtaining segmentation features that incorporate high-level semantic abstract features and low-level semantic detail features. Segmentation features provide pre-guidance and key information expansion for S²MF-CN. S²MF-CN shares the encoder and decoder parameters of SMF-SN, which enables S²MF-CN to obtain rich classification features. S²MF-CN uses the proposed Intra and Inter Mutual Guidance Attention Fusion (I²MGAF) module to first guide segmentation and classification feature fusion to extract hidden multi-scale contextual information. I²MGAF then guides the multidimensional fusion of genetic data and CT image data to compensate for the lack of information in single modality data. S²MMAM achieved 83.27% AUC and 81.67% accuracy in predicting KRAS gene mutation status in NSCLC. This method uses medical image CT and genetic data to effectively improve the accuracy of predicting KRAS gene mutation status in NSCLC.

available on a GitHub repository at https://github.com/xyttttboom/SSMMAM.

**Funding:** This work was supported by the National Natural Science Foundation of China (Grant No. U21A20469); the National Natural Science Foundation of China (Grant No. 61972274); the Central Government Guides Local Science and Technology Development Fund Project (Grant No. YDZJSX2022C004); the Natural Science Foundation of Shanxi Province (Grant No. 202103021224066); and NHC Key Laboratory of Pneumoconiosis Shanxi China Project, (Grant No.2020-PT320-005), the Non-profit Central Research Institute Fund of Chinese Academy of Medical Science. The funders had a role in decision to publish and preparation of the manuscript.

**Competing interests:** The authors have declared that no competing interests exist.

## Introduction

Lung cancer is specifically divided into non-small cell lung cancer (NSCLC) and small cell lung cancer. NSCLC accounts for approximately 85% of newly diagnosed lung cancers yearly [1]. The emergence of targeted therapy has substantially increased the survival rate of NSCLC patients. Prior to targeted therapy, it should be determined whether important disease-causing genes are mutated. KRAS is a common causative gene in NSCLC, and approximately one-third of patients with NSCLC have KRAS mutations. The usual diagnostic tool is a puncture biopsy. However, this invasive method has many limitations, such as it is unsuitable for all body types and has unpredictable consequences such as increased risk of cancer metastasis [2]. Therefore, there is an urgent need for a non-invasive diagnostic method that can accurately predict KRAS mutations in lung cancer patients. This method will not only improve the treatment outcome of patients but also guide prognosis.

In recent years, researchers have used CT images to predict gene mutations based on traditional radiomics and machine learning. Song et al. [3] propose a machine-learning model for predicting EGFR and KRAS mutation status. They used the model to extract statistical, shape, pathological, and deep learning features from 144 CT scans of tumor regions. Shiri et al. [4] used minimum redundancy, maximum correlation feature selection, and random forest classifier to build a multivariate model. The model analyzed radiological features extracted from images of tumors and successfully predicted EGFR and KRAS mutation status in cancer patients.

The radiomics and machine learning methods mentioned above have successfully predicted gene mutations. However, most of these methods rely on hand-crafted features. In recent years, deep learning based on convolutional neural networks has attracted much attention in the field of medical image computing. This data-driven approach can automatically extract complex image features [5–7]. In addition, imaging genomics is more expected to develop in the field of deep learning than single modality data for analytical studies. It integrates disease imaging data and genomic data. Imaging genomics is a high-throughput research method correlating imaging features with genomic data. In recent imaging genomics studies, researchers have proposed a series of deep learning algorithms and theoretical models based on image or genetic data. Dong et al. [8] proposed a multichannel and multitasking deep learning (MMDL) model. They used the fusion of radiological features of CT images and clinical information of patients to improve the accuracy of the model to predict KRAS gene mutations. Hou et al. [9] proposed a multimodal information fusion module based on attention that successfully predicted lymph node metastasis using deep learning features of CT images fused with genetic data. Therefore, machine learning and deep learning-based imaging genomics approaches have great potential and application in predicting KRAS gene mutation status in NSCLC.

Although the above model achieved considerable performance, there are still some challenges in the study of deep learning methods based on image and genetic data for predicting KRAS mutation status in NSCLC: 1) Majority of deep learning methods [8, 9] that study classification tasks focus only on classification methods. However, these studies did not use the segmentation features generated by the segmentation task to facilitate the classification task to improve the performance and effectiveness of the classification task. Lesion segmentation and classification are two highly related tasks. The segmentation can help remove distractions from CT images and thus is highly beneficial for improving the accuracy of lesion classification. 2) Most of the studied fusion methods used simple fusion means of direct concatenation. However, they ignore the correlation and difference between medical images and genetic data. It not only leads to ineffective mining of useful semantic features between

multi-scale image features and gene features but also fails to make full use of the complementarity of multimodal information. 3) Many studies used models that overemphasized the deep features of lesion abstraction. Nonetheless, they did not pay sufficient attention to the importance of detailed shallow features in prediction results. This leads to limitations in improving accuracy.

To overcome these difficulties and achieve non-invasive and accurate prediction of KRAS gene mutations in NSCLC. We propose a Semi-supervised Multimodal Multiscale Attention Model ($S^2$MMAM) for predicting KRAS gene mutation status in NSCLC. The model uses the Mean Teacher [10] framework as the main structure of the network. Mean Teacher can make full use of labeled images to achieve analytical prediction of unlabeled images in order to diminish the dependence of the network on manual annotation. In order to compensate for the information loss of single-modal unlabeled image data to the network, the model not only uses Semi-supervised Multimodal Fusion Classification Networks ($S^2$MF-CN) to share the parameter strategy of the Supervised Multilevel Fusion Segmentation Network (SMF-SN) to enrich the key information of the lesion. $S^2$MMAM also multi-modally fuses the patient's genetic data with the image data to expand the mutation knowledge. Specifically, SMF-SN designs a new Triple Attention-guided Feature Aggregation (TAFA) module. It aims to adaptively fuse high-level semantic features with low-level semantic features using an attention-guided mechanism. TAFA can ignore background noise and localize the extraction of lesion key features. In $S^2$MF-CN, we propose an Intra and Inter Mutual Guidance Attention Fusion ($I^2$MGAF) module to guide the fusion between inter-information and between intra-information in a staged manner. $I^2$MGAF can effectively extract complementary information from different modalities at different scales to facilitate classification efficiency improvement.

In contrast to conventional radiomics and machine learning [3, 4], we used a convolutional neural network technique for CT image feature extraction as compared to previous studies for KRAS mutation prediction. This technique is more efficient and reduces the cost of manual annotation. Moreover, it can realize the prospect of end-to-end applications. Studies [5–9] that have made predictions for other diseases in multimodal-based classification tasks have used simple multimodal fusion methods. In contrast, our proposed method focuses more on extracting different dimensions of information from different modal data to achieve complementary fusion.

The contributions of this paper are as follows:

- A Semi-supervised Multimodal Multiscale Attention Model ($S^2$MMAM) based on imaging genomics is proposed, which effectively solves the problem of difficult intermediate fusion of multimodal heterogeneous data. $S^2$MMAM exploits the facilitation of supervised segmentation features for semi-supervised classification tasks to improve the model performance for predicting KRAS gene mutation status in NSCLC.

- A new Triple Attention-guided Feature Aggregation (TAFA) module is designed. It is based on the attention module to adaptively fuse high-level semantic features with low-level semantic features. TAFA can suppress low-level background noise and retain detailed local semantic information.

- We use the Intra and Inter Mutual Guidance Attention Fusion ($I^2$MGAF) module to guide segmentation and classification feature fusion, as well as CT image and genetic data fusion, respectively. It can achieve multi-scale multimodal information fusion and improve classification performance.

## Related work

### Mean Teacher in semi-supervised learning

Semi-supervised learning has been studied in the medical imaging community for a long time [11, 12]. It can reduce the human workload on labeled data. Current research has shown the potential to improve network performance when labels are scarce. There are three semi-supervised models based on the principle of consistency: the $\Pi$-Model [13], Temporal Ensembling (TE) [13], and the Mean Teacher model. In order to show the advantages and disadvantages of three consistency-based semi-supervised methods more succinctly, we summarize Table 1, which allows a more precise comparison of the three approaches.

In recent years, Mean Teacher has achieved good results as a basic framework in semi-supervised classification tasks. Wang et al. [14] successfully identified diabetic macular edema based on the Mean Teacher model using a small amount of roughly labeled data and a large amount of unlabeled data. Liu et al. [15] used the Mean Teacher-based framework of the network model to successfully achieve skin lesion diagnosis with ISIC 2018 challenge and thorax disease classification with ChestX-ray14. Wang et al. [16] proposed a model that unifies diverse knowledge into a generic knowledge distillation framework for skin disease classification. It enables the student model to acquire richer knowledge from the faculty model. The above model demonstrates that Mean Teacher achieves excellent results in semi-supervised classification tasks, so we use it as the basic framework for Our $S^2$MMAM.

### Segmentation facilitates classification

Using segmentation tasks to facilitate classification network tasks is a basic form of multitask learning [17]. In multitask learning, the segmentation task associated with the classification task can assist the learning of the target by the classification task, thus improving the performance of the classification task [18]. Similarly, in a single-task classification model, this idea is borrowed from above. The information captured by the segmentation branch of the model can be transferred to the classification model to expand the foci information. The supervised segmentation task is trained using masked labeled data. The aim is to obtain the most comprehensive high-level semantic features of the target region and reduce the learning of noisy backgrounds. Rich segmentation features can support the classification task to learn more and richer semantic information. Thus, a supervised segmentation network can assist the classification task by suppressing the background noise introduced by missing physician labeling information in semi-supervised classification networks and improving the classification accuracy.

According to Table 2, the above works demonstrate that segmentation has a facilitating effect on classification. However, there is a common problem: they are all studied for supervised models. Supervised models have high requirements for data labeling costs. We believe

**Table 1. Comparison of three commonly used consistency-based semi-supervised methods.**

| Methods | Purpose | Limitations |
|---------|---------|-------------|
| $\Pi$-Model | Based on the consistency principle and perturbs the input data | High complexity and nosiy-prone results |
| Temporal Ensembling (TE) | Employs an exponential moving average (EMA) prediction for each unlabeled data as the consistency target. | Maintain a huge prediction matrix during the prediction process, and the training time complexity is high for large data sets. |
| Mean Teacher | Improves the problem of high time complexity caused by the TE method. Constructs a teacher model using the EMA weights of the student model. | |

**Table 2. Comparison of three commonly used consistency-based semi-supervised methods.**

| Methods | Contributions | Limitations |
|---|---|---|
| Xie et al. [18] | Proposed the Mutual Bootstrapping Deep Convolutional Neural Networks (MB-DCNN) model for simultaneous segmentation and classification of skin lesions. The rough lesion masks generated by the segmentation network in MB-DCNN help the classification network for training. The segmentation and classification networks transfer knowledge to each other in a bootstrap manner and facilitate each other. | 1. Non-end-to-end model<br>2. Professional doctors are needed to manually label each image |
| Zhao et al. [19] | Proposed a Segmentation-based Sequence Residual Attention Model (SSRAM) for the dual task of colorectal cancer lesion segmentation and KRAS gene mutation status prediction. The SSRAM utilizes the information provided by the segmentation network and the mask to successfully improve the accuracy of the classification task. | 1. Data pre-processing is more complex<br>2. Professional doctors are needed to manually label each image |
| Song et al. [20] | Utilized the lung nodule segmentation task to assist the lung nodule malignant development prediction task. | Professional doctors are needed to manually label each image |

that the combination of segmentation and classification tasks can make the network more informative. Therefore, our research aims to combine the idea of segmentation facilitating classification with semi-supervised models. We combined two related tasks of NSCLC lesion segmentation and KRAS gene mutation status prediction. $S^2$MMAM allows $S^2$MF-CN to obtain the key features of lesions upon initialization through the strategy of sharing network parameters between SMF-SN and $S^2$MF-CN. In $S^2$MF-CN, the segmentation features are guided to merge with the classification features to obtain the extracted key features. This strategy can enrich the lesion information and improve the network model classification performance.

## Multiscale features and attention learning

Traditional convolution operations mostly focus on extracting local features. However, due to the limited information contained in local features, the model cannot learn the full range of region of interest contents well. Multi-scale features contain local features of multiple regions of interest. The extracted local features are fused with other operations to obtain comprehensive information about the target, which helps the network model to learn. To extract multi-scale features, The Atrous Spatial Pyramid Pooling (ASPP) module [21] captures contextual information by multi-step convolution of the target region using different expansion rates. In the medical image domain, the PSE [22] module uses a patch-level pyramid design to extend SE operations to multiple scales, allowing the network to adaptively focus on vessels of variable width. The scale-aware Feature Aggregation (SFA) module [23] effectively extracts hidden multi-scale background information and aggregates multi-scale features to improve the model's ability to handle complex vasculature.

The Convolutional Block Attention Module (CBAM) [24] introduces channel and spatial attention. It extracts multiple key feature information from both dimensions to enrich the network content. In the medical image application domain, Context-assisted full Attention Network (CAN) [25] combines Non-Local Attention (NLA), Channel Attention (CA), and Dual-pathway Spatial Attention (DSA) to extract lesion information in multiple directions.

Currently, it is widely believed that both multi-scale features and attention mechanisms can help models enhance the recognition of feature maps from different dimensions. However, the above papers have a common problem: they do not combine the ideas of multi-scale and attention mechanism. Therefore, we combine these two techniques and design the TAFA module.

On the one hand, fuse high and low dimensional segmentation features to obtain abstract and detailed information. On the other hand, we fuse segmentation and classification features of different levels to guide the features to learn key factors adaptively and enhance the ability of the network to capture lesions. Thus, the predictive capability of the model is improved.

## Method

### Overview

In this paper, we propose a Semi-supervised Multimodal Multiscale Attention Model ($S^2$MMAM). The overall architecture of the model is divided into two parts: Supervised Multi-level Fusion Segmentation Network (SMF-SN) and Semi-supervised Multimodal Fusion Classification Network ($S^2$MF-CN), as shown in Fig 1. In this model, the useful information of CT images is captured by SMF-SN and transferred to $S^2$MF-CN to facilitate the execution of image prediction tasks. The $S^2$MMAM utilizes the fusion of CT images and genetic data to accurately predict whether KRAS is mutated in NSCLC.

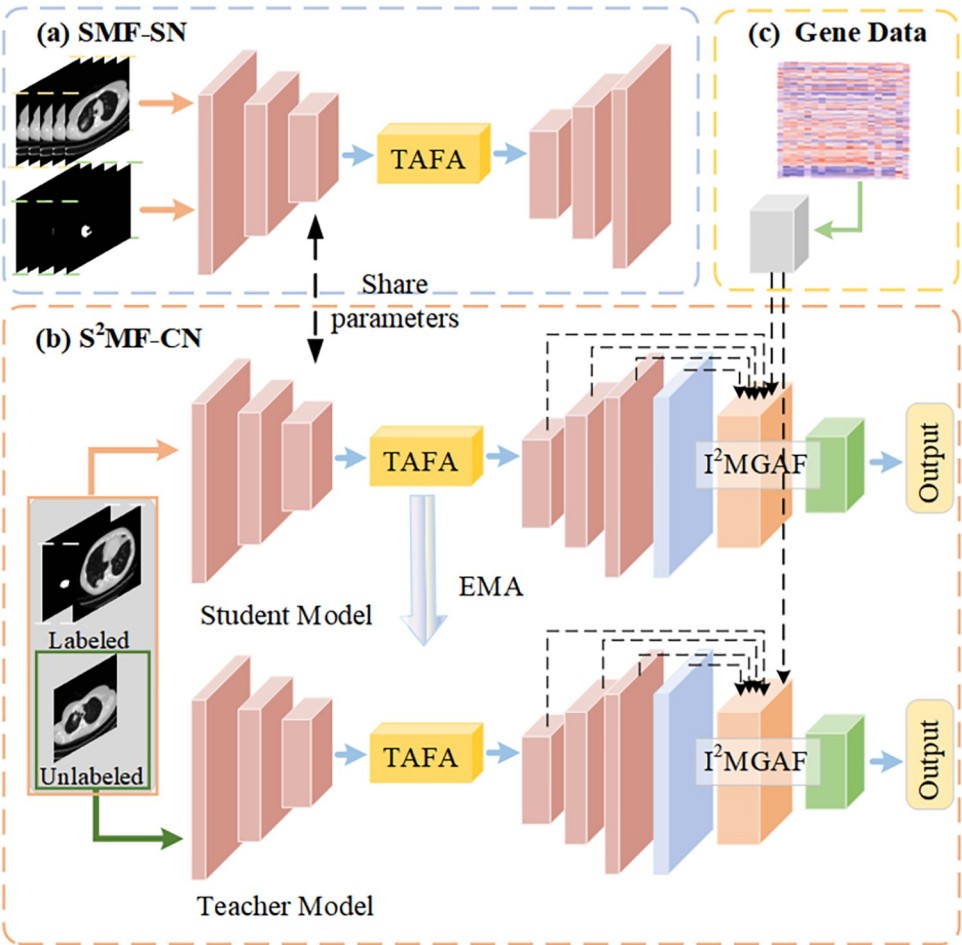

**Fig 1.** Overview of our $S^2$MMAM, including: (a) Supervised Multilevel Fusion Segmentation Network (SMF-SN). The inputs are CT images and pixel-level mask images, and the outputs are segmented lesion images, (b) Semi-supervised Multimodal Fusion Classification Network ($S^2$MF-CN), and (c) processing of gene data. In the $S^2$MMAM, the useful information of CT images is captured by SMF-SN and transferred to $S^2$MF-CN to facilitate the execution of image prediction tasks. The $S^2$MMAM utilizes the fusion of CT images and genetic data to accurately predict whether KRAS is mutated in NSCLC.

In the NSCLC dataset, each patient corresponds to a set of CT images and gene data (Section Dataset). Specifically, in our problem setting, we are given a training set containing $N$ labeled data and $M$ unlabeled data where $N<<M$. Let the labeled training dataset be denoted by $S_L = \{X_L^i, Y_L^i\}_{i=1}^N$ and $C_L = \{X_L^i, Y_L^i, Z_L^i\}_{i=1}^N$, where $S_L$ represents dataset for segmentation, $C_L$ represents dataset for classification, $X_L^i$ represents $i$-th labeled CT image, $Y_L^i$ represents the pixel-level annotation of $X_L^i$ and $Z_L^i$ represents the results of whether the KRAS gene is mutated. $Z_L^i \in \{0, 1\}$ where $0$ means negative and $1$ means positive. Let the unlabeled training dataset be denoted by $C_U = \{X_U^i\}_{i=1}^M$, where $X_U^i$ represents $i$-th unlabeled image. The entire model pipline can be summarized as follows: First, we pre-train SMF-SN, which is initialized on $S_L$, to train the network's ability to capture focal regions. It can eliminate problems such as large noise from CT and promote the ability of classification—meanwhile, the network body of S$^2$MF-CN shares encoder and decoder parameters with SMF-SN. Therefore, the encoder and decoder of S$^2$MF-CN are also initialized in this step, and practical segmentation features for different levels of lesions are obtained. The classification network in S$^2$MF-CN can capture the key classification features of lesions using these segmentation features. Finally, after S$^2$MF-CN fuses segmentation, classification, and genetic data features, the semi-supervised Student Model is trained to determine patients' KRAS gene mutation status accurately.

## Supervised multilevel fusion segmentation network

**The architecture of SMF-SN.**   This section introduces a supervised segmentation network based on multidimensional feature fusion. SMF-SN can precisely localize lesion edges and internal regions and greatly reduce the impact of image background noise on network performance. SMF-SN mainly utilizes our proposed SE-ResNeXt and TAFA modules.

We use the enhanced segmentation training dataset $S_L$ to train SMF-SN to obtain rich segmentation features. The obtained segmentation features can provide the semi-supervised classification network with a priori information about the lesion location. This improves the classification network's ability to localize and identify lesions.

As shown in Fig 2, SMF-SN includes a stem block, three encoder blocks, three TAFA blocks, a bridge block, three decoder blocks, and an output block.

In the encoder, each encoder is composed of a SE-ResNeXt and a max-pooling layer with step size 2. As shown in Fig 3, SE-ResNeXt is improved from ResNeXt with SENet. ResNext achieves aggregating a set of transitions with the same topology by repeating multiple blocks. SENet can perform feature learning on the aggregated features in the channel dimension to form the importance of each channel. SE-ResNeXt can enhance the network in both the channel and spatial dimensions to capture richer segmentation features. Applying the MaxPooling layer can reduce the spatial dimension of the feature map by half to reduce the computational cost. The output of the encoder is passed through a bridge consisting of SE-ResNeXt and Atrous Spatial Pyramid Pooling (ASPP). It provides the largest receptive domain for TAFA to include a wider range of contextual information, facilitating more efficient integration between multiple levels. Between high-level and low-level semantics, we use the proposed TAFA module. This module utilizes multi-scale and attention fusion mechanisms. The module both suppresses low-level irrelevant background noise and complements each other with contextual difference information, preserving more detailed local semantic information and better learning of focal information. TAFA module is depicted in detail in Section Triple Attention-guided Feature Aggregation.

**Triple attention-guided feature aggregation.**   Since CT images of lung nodules may contain a large amount of noise, for example, there are problems of grayscale overlap between lung tissues, blurred boundaries, and challenging to distinguish. High-level features of the decoder and low-level features of the encoder are crucial for capturing lesion features.

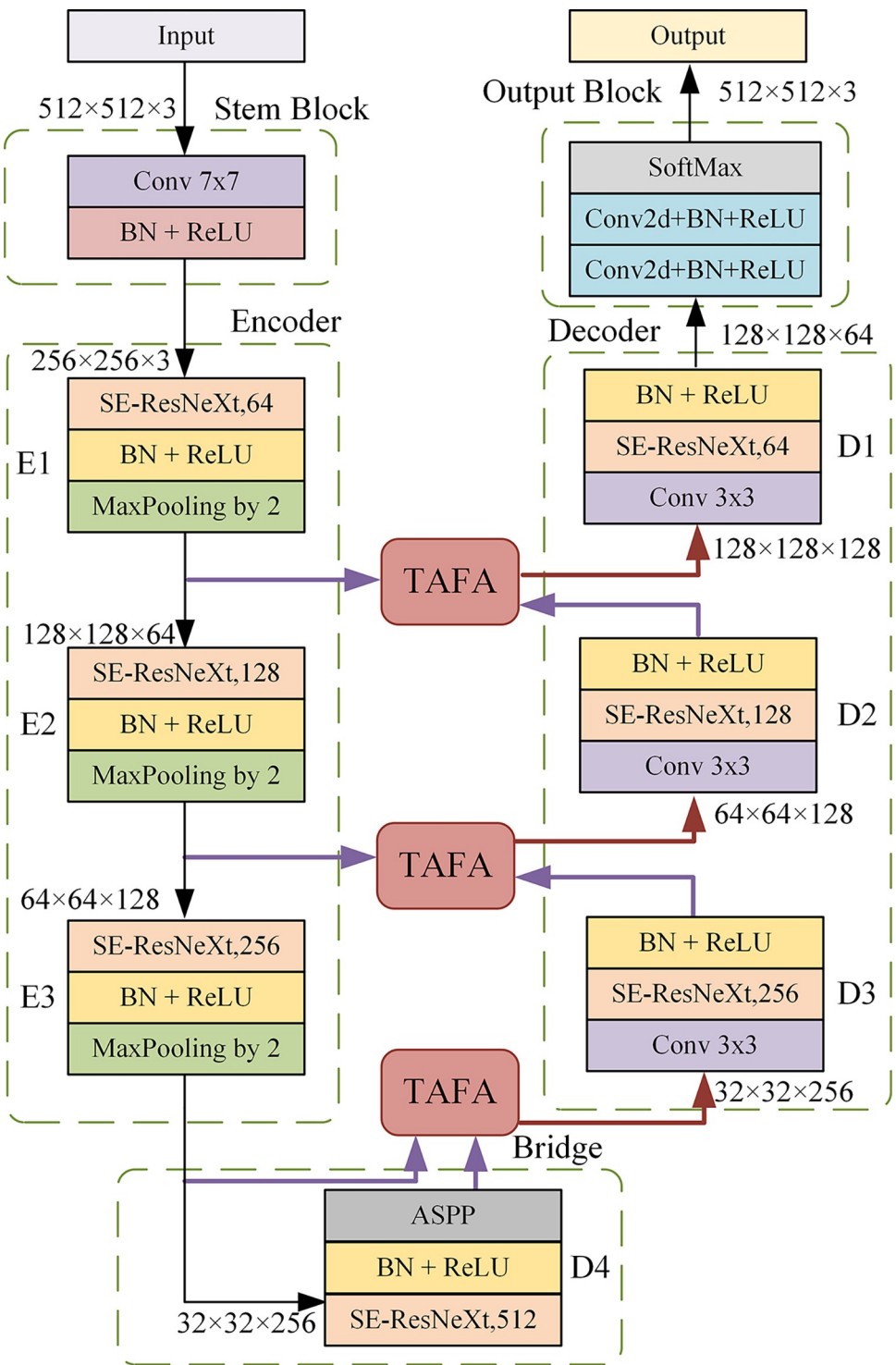

**Fig 2. Block diagram of the proposed SMF-SN architecture.** We adjust the dilation rates in ASPP in the bridge from 6,12,18 to 3,6,9 to better adapt SMF-SN to our segmentation task.

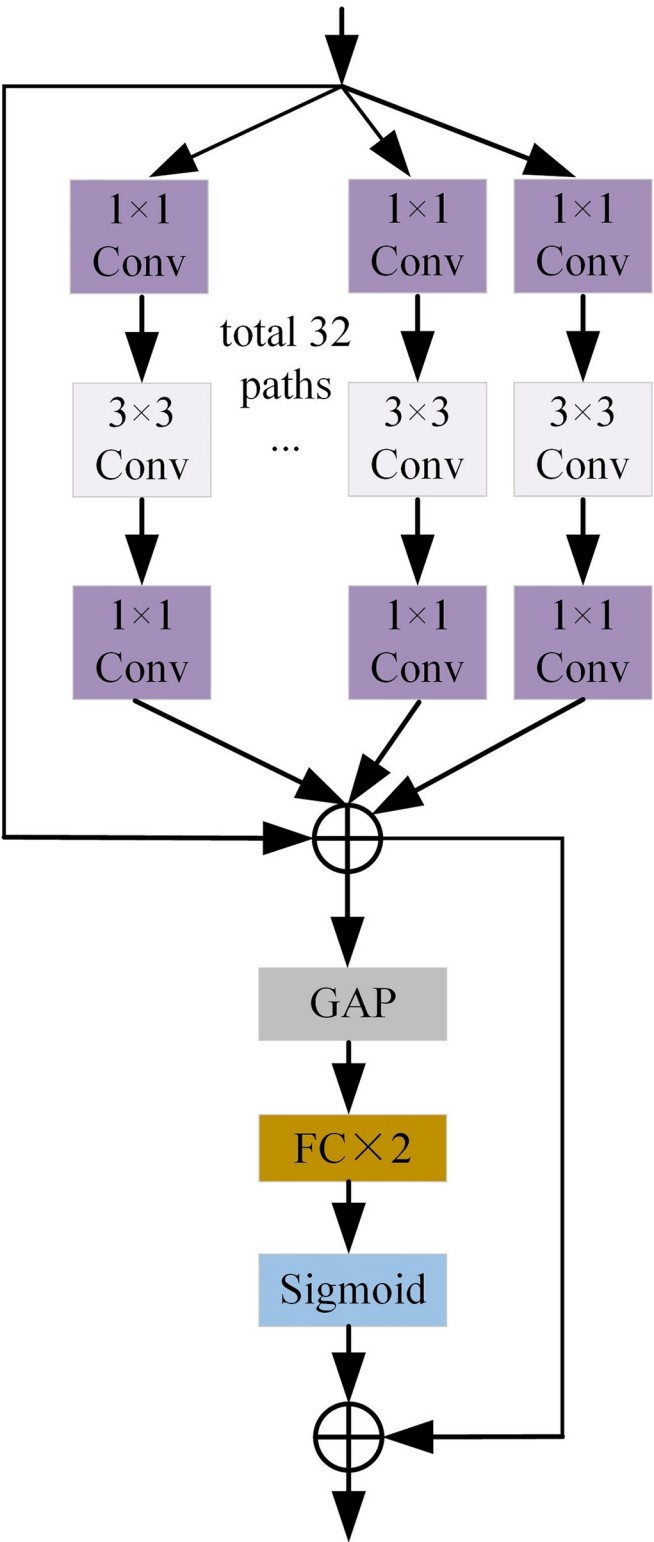

**Fig 3. The architecture of SE-ResNext.** SE-ResNeXt is improved from ResNeXt with SENet.

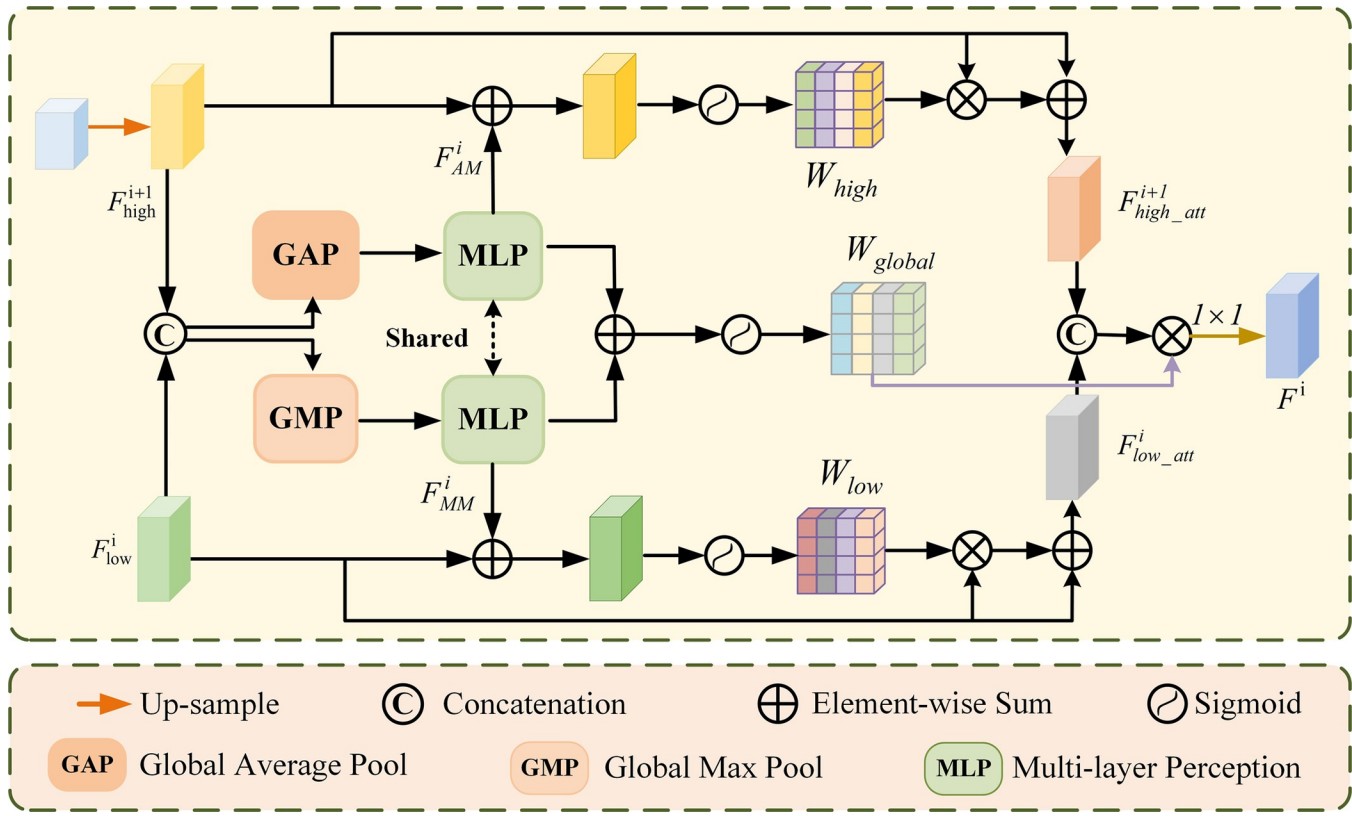

**Fig 4. Framework diagram of the proposed TAFA module.**

However, most of the existing UNet-based connection methods directly connect shallow and deep semantic features of different scales. This behavior ignores that high-level features contain rich semantic information that can help low-level features identify semantically important locations. Likewise, low-level features contain rich spatial information that can help high-level features reconstruct accurate details.

Considering the above factors, we design a Triple attention-guided feature aggregation (TAFA) module to guide the fusion between high and low-dimensional features. TAFA can guide different layers to extract key feature information individually and then fuse after retaining the domain invariant key information, as shown in Fig 4. In the TAFA module, we first upsample the high-dimensional feature $F_{high}^{i+1}$ to have the same size as the low-dimensional feature $F_{low}^{i}$ ($i \in \{1, 2, 3\}$). After that, we perform the high and low-dimensional feature concatenating based on channels to obtain $F_C^i$.

$$F_C^i = Concat(F_{low}^i, f_{up}(F_{high}^{i+1}))  \qquad (1)$$

Where *Concat* represents the concatenation operation, $f_{up}$ represents up-sampling operations. Then, to better mine the most useful feature channels between different levels. We introduce a scale channel attention-aware mechanism to automatically select the appropriate receptive domain for the feature map and suppress the interference of irrelevant background noise. We feed the concatenate feature $F_C^i$ of high and low dimensional features into global average pooling (GAP) and global max pooling (GMP) respectively. TAFA uses the GAP module to excite the feature channel information and the GMP layer to retain the semantic

maximum information. Afterward, the corresponding feature maps $F_{AM}^i$ and $F_{MM}^i$ are obtained using a multi-layer perceptron (MLP) sharing the same parameters. The feature maps $F_{AM}^i$ and $F_{MM}^i$ are summed. Then the sum feature passes through a sigmoid function to generate a global bootstrap feature coefficient $W_{global}$.

$$W_{\text{global}} = f_\sigma \{ f_{mlp}(f_{gmp}(F_C^i)) \oplus f_{mlp}(f_{gap}(F_C^i)) \} \tag{2}$$

Where $f_\sigma$ represents sigmoid activation, $f_{mlp}$ represents the MLP operator, $f_{gap}$ represents the global average pooling, $f_{gmp}$ represents the global max pooling. In addition, using the high and low level semantic binding information $F_{AM}^i$ and $F_{MM}^i$ as guidance, they are combined with high and low dimensional features, respectively, and the high level guidance semantic features $F_{high\_att}^{i+1}$ and low level guidance semantic features $F_{low\_att}^i$ are obtained after the attention operation, respectively.

$$W_{high} = f_\sigma(F_{high}^{i+1} \oplus F_{AM}^i), \, W_{low} = f_\sigma(F_{low}^i \oplus F_{MM}^i) \tag{3}$$

$$F_{high\_att}^{i+1} = F_{high}^{i+1} \otimes W_{high} \oplus F_{high}^{i+1}, \, F_{low\_att}^i = F_{low}^i \otimes W_{low} \oplus F_{low}^i \tag{4}$$

Finally, the weighted features are concatenated. The concatenated feature maps are multiplied with $W_{global}$. Then domain-invariant information is captured while reducing the dimensionality through 1x1 convolutional layers to obtain the final fusion module $F^i$.

$$F^i = Conv(W_{global} \otimes (Concat(F_{low\_att}^i, F_{high\_att}^{i+1}))) \tag{5}$$

Where $Conv$ represents $1{\times}1$ convolution operation, $\oplus$ represents element-wise sum and $\otimes$ represents element-wise multiplication.

Our proposed TAFA transfers features from shallower convolutional layers to deeper convolutional layers. Performing the shallow features in the deeper convolutional layers prevents the shallow features from being forgotten. It makes the obtained features have more vital characterization ability. By gradually guiding the fusion between high and low features, SMF-SN can be guided to adaptively combine high and low-dimensional semantic information to reassign feature weights and better capture critical domain invariant information. Thus, lung nodules can be separated from the noise.

## Semi-supervised multimodal fusion classification networks

**The architecture of S²MF-CN.** The proposed S²MF-CN structure is shown in Fig 1(B), which adopts the Mean Teacher model structure as the main framework of the classification network. In Mean Teacher, the Teacher network has the same structure as the Student network. The Student model is the target model to be trained. It assigns the exponential moving average (EMA) of its weights to the Teacher model at each step of training. The predictions of the Teacher model will be considered as additional supervision of the learning of the Student model. Our model uses the final Student model to make predictions. The specific training Student model is shown in Fig 5(A) and consists of three parts: encoder, decoder, and Intra and Inter Mutual Guidance Attention Fusion (I²MGAF) Module. The encoder and decoder have the same structure and parameters as the SMF-SN. This allows focusing on the lesion region and capturing the necessary segmentation features through the encoder and decoder. I²MGAF performs feature purification using mutual guidance attention modules. It is able to extract multi-scale lung CT image features and genetic features fully. It also performs an adaptive fusion of features through an attentional fusion mechanism for KRAS gene mutation

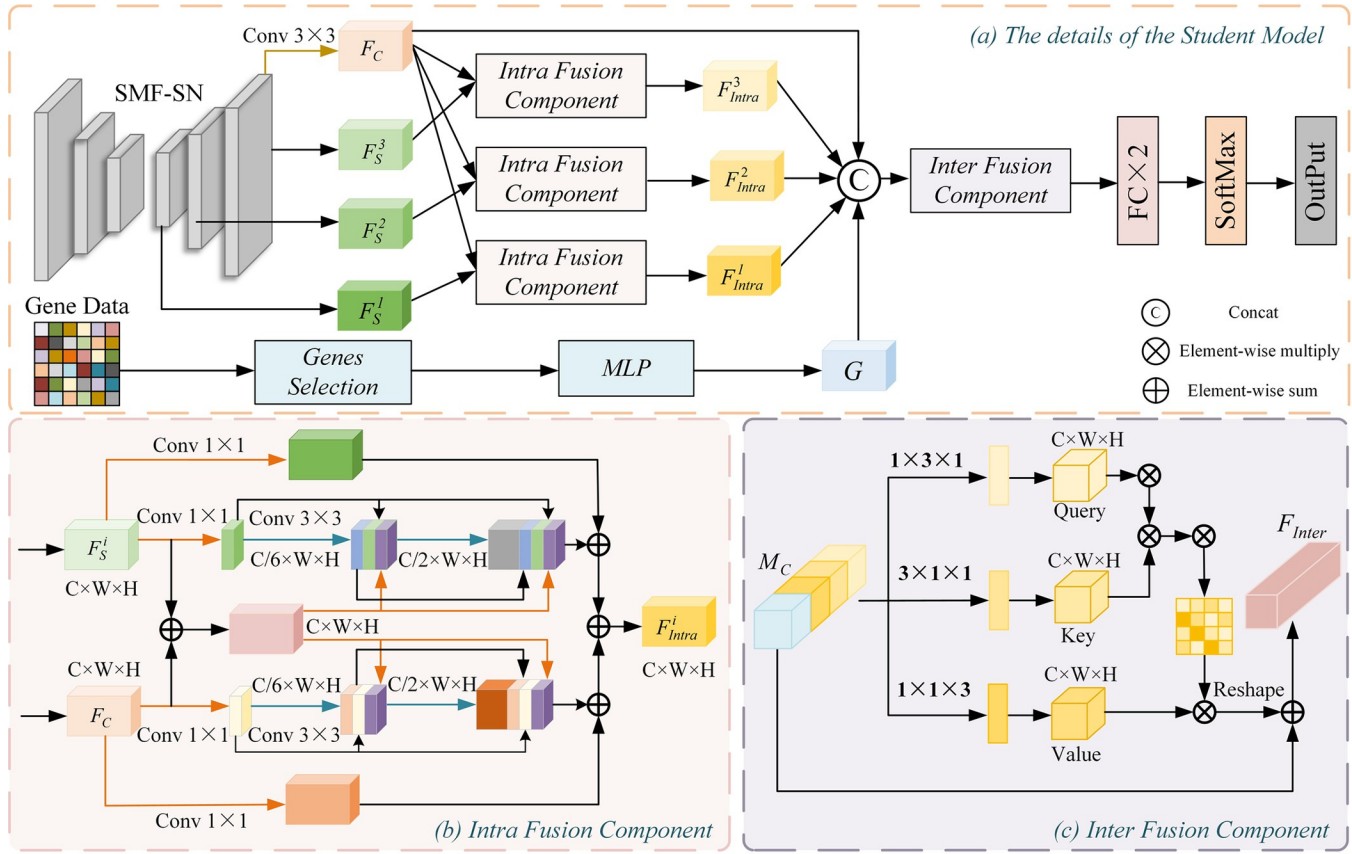

**Fig 5.** The overview of the Student Module, including (a) the specific implementation details of the Student Model, (b) Intra fusion component (IntraFC) aims to fuse classification and segmentation features at different levels, and (c) Inter fusion component (InterFC) aims to fuse CT image features and genetic features.

prediction in NSCLC. I$^2$MGAF is described in detail in Section Intra and Inter Mutual Guidance Attention Fusion Module.

**Intra and inter mutual guidance attention fusion module.** In the S$^2$MF-CN network, we propose an I$^2$MGAF module. I$^2$MGAF fully fuses multi-scale image segmentation, classification features, and genetic features by using the IntraFC component and InterFC component with a dual attention fusion mechanism. Its aim is to improve the classification capability of the classification network.

1. Intra Fusion Component (IntraFC)

   We propose the IntraFC based on the MultiRes Block, which can capture multi-scale information [26]. We adopted a strategy of fusing classification features with segmentation features at each level. The information favoring the prediction of KRAS gene mutation status is jointly retained.

   The specific structure of the IntraFC component is shown in Fig 5(B). The final level segmentation features $F_S^3$ are subjected to convolutional operations to obtain the initial classification features $F_C$. Due to the problem of induction bias inherent in the convolution mechanism, it is easy to lose the key features of the lesion after multiple convolutions. Therefore, it is necessary for us to fuse the previous segmentation features with the existing classification features to compensate for the bias problem due to the deep network. First we reshape the segmented feature $F_S^i\{i \in (1, 2, 3)\}$ through dimensionality until $C \times W \times H$ is

the same size as the classified feature $F_C$. Then, after the segmentation features and classification features are each applied $3 \times 3$ convolution. We will introduce the convolutional features from the previous stage and the initial fusion feature $F_{SC}^i$ before the subsequent convolution. This can effectively model the correlation between segmentation and classification features. It ensures that the features from the shallow convolutional layer of segmentation and classification are better transferred to the deeper layers. The final fused result $F_{Intra}^i \{i \in (1, 2, 3)\}$ is obtained after several feature fusions.

2. Inter Fusion Component (InterFC)

We propose the InterFC to find the bidirectional mapping relationship between lung cancer image features and causative genes from the sagittal view (x-axis), coronal view (y-axis), and axial view (z-axis), respectively. InterFC can adaptively enhance the necessary information in different modal features, allowing a more adequate fusion of multimodal features. The specific structure of the InterFC component is shown in Fig 5(C). The initial classification feature $F_C$, the fusion result $F_{Intra}^i \{i \in (1, 2, 3)\}$ output by IntraFC, and the processed genetic data $G$ are firstly subjected to a splicing operation to obtain the multimodal fusion feature $M_C$. After that, $M_C$ is delivered to InterFC to further model the importance of each modal data.

$$M_C = Concat(F_{Intra}^1, F_{Intra}^2, F_{Intra}^3, F_C, G) \tag{6}$$

Where $Concat$ denotes the concatenation operation. Then the concatenated multimodal data features are fed to three convolutional layers with BN and ReLU. The size of the convolution kernel is $1 \times 3 \times 1$, $3 \times 1 \times 1$ and $1 \times 1 \times 3$ respectively, to produce three feature maps $Query \in R^{C \times H \times W}$, $Key \in R^{C \times H \times W}$ and $Value \in R^{C \times H \times W}$ (where $C, H, W$ indicate the channel, height, width of the input features $F$ respectively). We first transpose the Query feature. Then, we perform a softmax layer on the matrix multiplication of $Query^T$ and $Key$ to encode the feature relationships in sagittal and coronal views. Finally, matrix multiplication is multiplied with $Value$ to obtain the voxel-level attention enhanced fusion features $F_{Inter}$, which are then reshaped to be in $R^{C \times H \times W}$.

$$F_{Inter} = M_C \oplus soft\max(Query^T \otimes Key) \otimes Value \tag{7}$$

Where $\oplus$ denotes element-wise sum, $\otimes$ denotes element-wise multiplication.

## Data

### Dataset

In this study, we applied NSCLC-Radiogenomics [27], directly accessible on the Cancer Imaging Archive (TCIA) website. NSCLC-Radiogenomics is part of a public dataset. The patients involved in the dataset have been ethically approved. Users can download the relevant data for research and publication free of charge. Our study is based on open-source data and is therefore free from ethical issues and other conflicts of interest. NSCLC-Radiogenomics has developed a unique radiogenomic dataset from the NSCLC dataset of 211 subjects. The imaging data include mainly CT, semantic annotation of tumors observed on CT images using controlled vocabulary, and segmentation maps of tumor lesions (lung nodules) on CT scans; the genetic data include mainly RNA sequencing (RNA-seq) data. In the training and testing datasets, patients would be excluded for 1) lack of RNA-seq data, 2) lack of CT images, and 3) lack of physician-annotated segmentation maps of CT lesions. After screening, the number of cases with complete images and genetic data was 124. Of the 124 patients, 94 were of the wildtype,

**Table 3. Patients' medical record information in the dataset.**

| Category | Total | Mutation | Wildtype |
|---|---|---|---|
| **Amount** | 124 | 30 | 94 |
| **Gender** | | | |
| Male | 93 | 24 | 69 |
| Female | 31 | 6 | 25 |
| **Smoking History** | | | |
| Smoking | 107 | 30 | 77 |
| Non-smoking | 17 | 0 | 17 |
| Pathological type | | | |
| Adenocarcinoma | 105 | 29 | 76 |
| Squamous Carcinoma | 17 | 0 | 17 |
| Other | 2 | 1 | 1 |

and 30 were of the mutation type. The clinical information of these patients is shown in Table 3. All data were randomly divided into training and test datasets in a 4:1 ratio.

## Data preprocessing

**CT image.** In our experiments, for 124 sets of CT images inspired by Cubuk et al. [28], we use the simple procedure of AutoAugment to automatically search for improved data enhancement strategies. By designing a search space in which a strategy consists of many sub-strategies, one sub-strategy is randomly selected for each image in each small batch. The sub-strategies contain two operations, each of which is an image processing function, such as clipping or applying the probability and magnitude of that function. Thus, we obtained 6696 images with a fixed size of 512×512.

**Genes selection.** The gene expression data used in this study is RNA-seq data. Since the vast gene dataset contains more than 20,000 gene expression data per patient, the huge amount of gene expression data can significantly increase the computational cost and decrease the prediction accuracy. Therefore, before training the model, we screened the gene expression data from RNA-seq sequencing by the feature selection algorithm [29] to retain the most relevant genes with KRAS mutations. A total of 115 relevant genes were finally screened. The obtained correlated genes were fed into MLP to obtain effective gene features, which achieved mapping high-dimensional gene data to low-dimensional space.

## Experiments and results

### Implementation details

Our model S$^2$MMAM is divided into SMF-SN and S$^2$MF-CN. The labeled image data applied to SMF-SN is 30% of the total dataset, about 2100 images. The training dataset applied to S$^2$MF-CN consists of 30% labeled data and 70% unlabeled data. Our experiments are mainly done on 2 NVIDIA RTX A5000 GPUs and 64 GB of memory. All models in the experiments are trained using 10-fold cross-validation. The specific initialization network configurations are shown in Table 4.

### Evaluation metrics

To quantitatively analyze the experimental results, we used six performance metrics to evaluate the classification results obtained, including Accuracy (AC), Recall, Precision, Specificity (SP), Area Under the receiver operating Curve (AUC) and F1 score (F1). They are defined as

**Table 4. The initialization network configurations of model.**

| Network Configurations | Setting |
|---|---|
| Epochs per fold | 20 |
| Optimizer | Adams |
| Initial learning rate | 0.001 |
| Batch size | 16 |

follows:

$$AC = \frac{TP + TN}{TP + TN + FN + FP} \tag{8}$$

$$Recall = \frac{TP}{TP + FN} \tag{9}$$

$$Precision = \frac{TP}{TP + FP} \tag{10}$$

$$SP = \frac{TN}{TN + FP} \tag{11}$$

$$AUC = \int_0^1 t_{pr}(f_{pr})df_{pr} = P(X1 > X0) \tag{12}$$

$$F1 = 2 \times \frac{Recall \times Precision}{Recall + Precision} \tag{13}$$

Where *TP* is true positive, *TN* is true negative, *FP* is false positive, *FN* is false negative, $t_{pr}$ is the true positive rate, $f_{pr}$ is the false positive rate, *X1* and *X0* are the confidence scores for negative instances of sexual instances, respectively.

## Ablation studies

In this section, we evaluate the impact of the SE-ResNeXt, TAFA module, and the I²MGAF module on our S²MMAM respectively.

**Ablation study of SE-ResNeXt.** Using SE-ResNeXt as the backbone of the network can not only enhance the network to extract focal features. It can also take advantage of the lightweight feature of ResNeXt to reduce the computational burden of the network and improve the network's efficiency. To verify the performance of our proposed SE-ResNeXt, we replace the backbone network with S²MMAM(UNet), S²MMAM(ResNet), S²MMAM(ResNeXt) and S²MMAM(Inception-V3), respectively. These methods compare with our proposed SE-ResNeXt on the same dataset. The results are shown in Table 5.

As shown in Table 5, it is evident from the results that our S²MMAM(Ours) performed the best in KRAS gene mutation prediction among the five models. S²MMAM(Ours) achieved the best results in all six comparative metrics. The AUC was 83.27%, 5.96% higher than the second-place S²MMAM(ResNeXt). Compared to the more popular S²MMAM (Inception-V3), the AUC was 6.43% higher. SE-ResNeXt has a simpler architecture and lower computational complexity than Inception-v3. SE-ResNeXt effectively eliminates the semantic differences

**Table 5. Comparison of classification performance of UNet, ResNet, ResNeXt, Inception-v3 and SE-ResNeXt on S$^2$MMAM.** SE-ResNeXt(Ours) achieved the best results in all six comparative metrics.

| Methods | AC(%) | Recall(%) | Precision(%) | SP(%) | AUC(%) | F1(%) |
|---|---|---|---|---|---|---|
| UNet [30] | 71.29±0.53 | 71.35±0.11 | 73.24±0.64 | 70.09±0.36 | 70.33±0.38 | 72.28±0.37 |
| ResNet [31] | 73.95±1.05 | 74.01±2.19 | 74.15±0.41 | 76.32±2.46 | 76.48±0.47 | 74.07±1.29 |
| ResNeXt [32] | 77.99±3.16 | 76.92±1.37 | 78.14±3.21 | 75.28±3.14 | 77.31±2.22 | 77.53±2.27 |
| Inception-v3 [33] | 75.29±2.86 | 77.39±3.15 | 75.14±2.21 | 77.20±2.18 | 76.84±1.43 | 76.25±2.66 |
| SE-ResNeXt(Ours) | **81.67±2.67** | **82.31±2.51** | **83.15±1.21** | **82.66±2.07** | **83.27±1.49** | **82.73±1.86** |

between features by utilizing multi-scale and attention mechanisms. This enables SE-ResNeXt to outperform other traditional networks trained on the data and helps the model to better localize the lesion area.

**Ablation study of TAFA module.** Using TAFA as the basic module to build S$^2$MMAM can better capture the key and complementary information of high-level semantic features and low-level semantic features. It further enhances the feature representation capability, improves the model to extract segmented feature quality and promotes classification performance. To validate the performance of our proposed TAFA, we compare our proposed S$^2$MMAM (Ours) with Addition, Concatenation, Adaptive Enhanced Attention Fusion (AEAF) [34], and Adaptive Spatiotemporal Semantic Calibration Module (ASSCM) [35] on the test dataset, respectively. The results are shown in Table 6.

The results show that the highest performance metrics were achieved on the classification task using our proposed S$^2$MMAM constructed from TAFA. TAFA (Ours) not only obtained the highest AUC value of 83.27% compared to the other four models. It also achieved the best results on the other five classification performance metrics, with a maximum AC of 81.67% and a maximum SP of 82.66%. The AUC is 4.39% higher compared to the second place AEAF, proving that TAFA can effectively fuse multi-scale information. It proves that our model S$^2$MMAM can better detect more patients and effectively reduce the underdiagnosis rate. TAFA achieved 82.73% in F1 score, which is higher than the AEAF at 4.46% and the ASSCM at 4.3%. It is demonstrated that our TAFA has a more stable classification performance and better classification ability.

**Ablation study of I$^2$MGAF module.** The I$^2$MGAF module was implemented to guide the fusion of features in segmentation and classification tasks, as well as the fusion of image features with genetic data. To demonstrate that the I$^2$MGAF module can better guide the fusion of multimodal and multiscale features in the model. We replaced the IntraFC module in I$^2$MGAF with Addition, Concatenation, and Adaptive Feature Fusion (AFF Block) [23], respectively. The InterFC module was replaced with Group Feature Learning (GFL Block) [36] and Non-Local Attention (NLA Block) [25], respectively. The five obtained models are compared with the performance of I$^2$MGAF on the classification test dataset. The results are shown in Figs 6 and 7.

**Table 6. Comparison of classification performance of TAFA on S$^2$MMAM and four models with different fusion blocks.** TAFA(Ours) achieved the best results in all six comparative metrics.

| Methods | AC(%) | Recall(%) | Precision(%) | SP(%) | AUC(%) | F1(%) |
|---|---|---|---|---|---|---|
| Addition | 72.56±1.02 | 72.28±1.69 | 71.43±2.14 | 72.47±1.79 | 72.49±1.23 | 71.85±1.97 |
| Concatenation | 73.03±1.52 | 73.63±1.46 | 75.24±1.22 | 73.13±1.86 | 72.15±0.87 | 74.43±1.34 |
| AEAF [34] | 77.25±1.67 | 78.22±1.62 | 77.73±2.44 | 76.56±2.77 | 78.88±2.45 | 78.27±2.72 |
| ASSCM [35] | 78.39±1.42 | 78.37±1.19 | 78.49±0.76 | 78.57±2.06 | 77.89±1.06 | 78.43±0.97 |
| TAFA(Ours) | **81.67±2.67** | **82.31±2.51** | **83.15±1.21** | **82.66±2.07** | **83.27±1.49** | **82.73±1.86** |

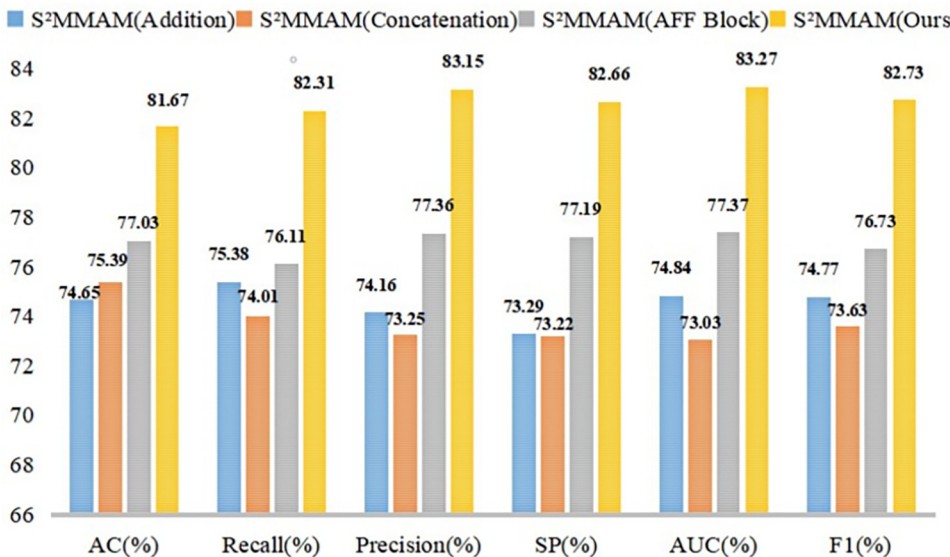

**Fig 6. Comparison of the classification performance of IntraFC and three models using other fusion methods.**

Fig 6 shows a visual comparison of the six classification performance metrics after replacing the IntraFC module in I²MGAF with addition, concatenation, and AFF Block, respectively. From Fig 6, we find that the Concatenation fusion method achieves the lowest AUC value, so the [5–9] method cannot fully take advantage of the multimodal information. AFF Block is 5.9% lower than our IntraFC in AUC. This is due to the fact that AFF Block only focuses on inter-channel fusion of features at different levels, ignoring the potential loss of information

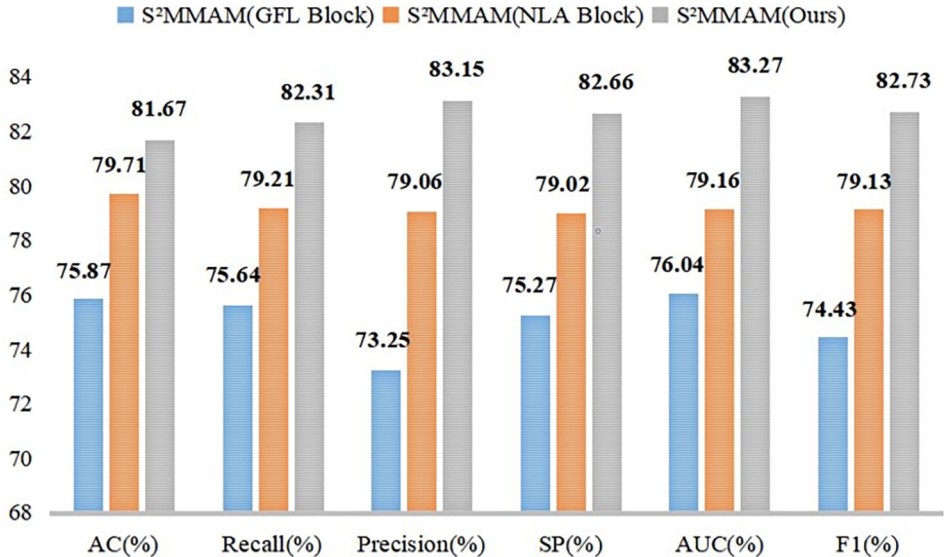

**Fig 7. Comparison of the classification performance of InterFC and two models using other fusion methods.**

due to network depth. Our IntraFC module not only focuses on channel fusion of segmentation and classification features but also solves the problem of information loss caused by multiple fusions.

Fig 7 shows the comparison of the six classification performance metrics after replacing the InterFC module in $I^2$MGAF with the GFL Block and NLA Block, respectively. Our InterFC outperforms the second-place NLA Block by 4.11% and 3.6% in AUC and F1 scores, respectively. Our InterFC solves the limitation that NLA Block only focuses on the fusion of information in a single dimension. InterFC can fully combine the information in three dimensions to fuse the data of different modalities and improve the model sensitivity, thus obtaining a better prediction of KARS mutation.

## Comparison experiment

We compare the proposed $S^2$MMAM with the classical Semi-supervised Learning (SSL), and the recently published SSL image classification models with better results, trained on data with 100% and 30% of labeled data, respectively. Among the classical SSL methods include Π-Model [13] and Mean Teacher. The competing methods include Relation-driven Self-ensembling Model (RSM) [15], SS-TBN [37], and DAB [38]. Note that we reproduce the above methods on the same testset for the sake of fairness.

Table 7 shows that the key evaluation metrics of $S^2$MMAM outperform the other models on both 100% and 30% of the data with labeled data. This means that our $S^2$MMAM can be used not only for supervised training but also for semi-supervised applications. We use the fully supervised model with 100% labeled data as the upper bound. And the SSL model trained on 30% labeled data as the target model. As can be seen from the Table 7, $S^2$MMAM(Ours) achieved an AUC of 83.27% on the 30% labeled dataset. Mean Teacher only obtains an AUC result of 80.04% on the 100% labeled dataset. This shows the superiority of our $S^2$MMAM for the classification task and even achieves accurate prediction with less cost. Compared with other models, our $S^2$MMAM has the smallest gap of AUC, which is only 4.65% between 30% of the labeled dataset and the upper bound. This result indicates that our TAFA module and $I^2$MGAF module effectively fuse the key features of multi-scale multi-modality. They can solve the problem of feature disappearance due to deep convolution and re-establish the fusion of high and low dimensional semantic key features. Compared with other SSL models that use only CT images for classification, our model has an AUC 6.9% higher than the second best

**Table 7. Comparison of the classification performance of $S^2$MMAM and five other semi-supervised medical image classification models.**

| Methods | Labeled | Unlabeled | Data | | Result(%) | | | | | |
|---|---|---|---|---|---|---|---|---|---|---|
| | | | CT | Gene | AC | Recall | Precision | SP | AUC | F1 |
| Π-Model [13] | 100% | 0 | √ | | 76.35±2.32 | 78.21±2.36 | 79.32±2.68 | 77.32±2.65 | 76.23±2.31 | 78.76±2.51 |
| Mean Teacher [10] | | | √ | | 81.24±2.18 | 80.15±1.81 | 82.34±1.79 | 81.86±2.43 | 80.04±2.78 | 81.23±1.8 |
| RSM [15] | | | √ | | 84.21±1.26 | 81.93±2.15 | 84.21±2.03 | 84.72±2.07 | 83.41±2.65 | 83.05±2.09 |
| SS-TBN [37] | | | √ | | 80.25±1.71 | 79.38±2.16 | 79.88±2.71 | 78.62±3.89 | 81.23±2.44 | 80.35±2.66 |
| DAB [38] | | | √ | | 81.79±2.3 | 80.42±1.51 | 82.11±2.23 | 82.8±2.43 | 83.56±2.36 | 82.37±1.97 |
| $S^2$MMAM(Ours) | | | √ | √ | **86.94±3.12** | **85.97±2.19** | **84.28±1.73** | **86.11±2.54** | **87.92±1.69** | **85.12±1.96** |
| Π-Model [13] | 30% | 70% | √ | | 71.23±2.49 | 72.11±1.65 | 72.56±1.24 | 71.16±2.77 | 70.15±2.17 | 72.33±1.45 |
| Mean Teacher [10] | | | √ | | 74.28±2.53 | 74.16±2.73 | 75.29±2.94 | 74.62±2.82 | 74.21±3.48 | 74.72±2.83 |
| RSM [15] | | | √ | | 75.91±2.37 | 75.13±3.21 | 76.37±2.86 | 75.49±3.53 | 75.94±2.34 | 75.74±3.04 |
| SS-TBN [37] | | | √ | | 76.01±1.54 | 75.17±1.89 | 77.22±1.47 | 77.01±2.15 | 76.37±2.22 | 77.74±1.98 |
| DAB [38] | | | √ | | 75.73±2.46 | 77.22±2.49 | 76.16±2.73 | 76.49±2.87 | 76.06±1.87 | 76.58±2.12 |
| $S^2$MMAM(Ours) | | | √ | √ | **81.67±2.67** | **82.31±2.51** | **83.15±1.21** | **82.66±2.07** | **83.27±1.49** | **82.73±1.86** |

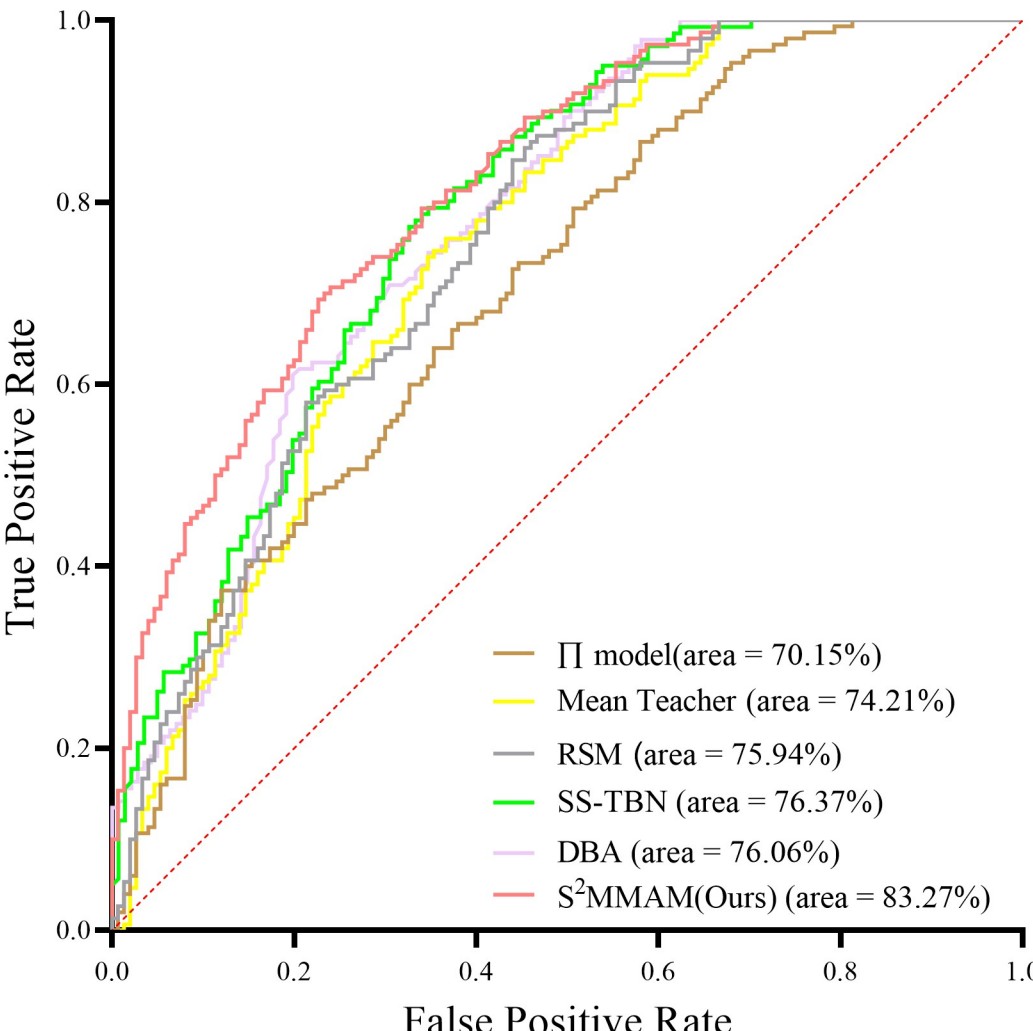

**Fig 8. AUC of our S²MMAM and five other medical image classification models on 30% labeled image dataset.**

SS-TBN model and 7.21% higher than the DBA model. This is due to our design of a new multimodal fusion module, I²MGAF. I²MGAF guides the fusion of features for multiple tasks and the fusion of multimodal data. It utilizes segmentation features to facilitate the classification task and efficiently extract important features from different modalities. I²MGAF has the ability to compensate for the specificity information that can be easily overlooked by a single data modality and achieve the complementary effects of multi-modal data. As well as to find the pathogenic features of lesions based on multi-dimensionality, thus enhancing the classification ability. We also plot the AUC curves of our S²MMAM with the other five models in Fig 8 to demonstrate the classification performance of our S²MMAM more visually.

## Discussion

### Superiority of the model

Although ablation studies and comparison experiments have demonstrated the merits of our proposed method, further discussions are needed on 1) the positive effects of segmentation

**Table 8. Six metrics were achieved on the test set by Baseline, Baseline+SMF-SN, Baseline+Gene, and our S²MMAM when using 30%, 40%, and 100% labeled training images.**

| Methods | Labeled | Unlabeled | Data | | Result(%) | | | | | |
|---|---|---|---|---|---|---|---|---|---|---|
| | | | CT | Gene | AC | Recall | Precision | SP | AUC | F1 |
| Baseline | 100% | 0 | √ | | 76.29±5.22 | 75.39±2.31 | 74.92±2.64 | 77.57±2.84 | 78.26±3.14 | 75.15±2.39 |
| Baseline+SMF-SN | | | √ | | 83.19±2.46 | 79.38±1.37 | 81.31±1.67 | 82.47±3.64 | 84.29±4.36 | 80.33±1.52 |
| Baseline+Gene | | | √ | √ | 82.37±1.67 | 80.06±3.49 | 78.15±2.17 | 81.66±3.58 | 82.2±2.61 | 79.09±2.81 |
| S²MMAM(Ours) | | | √ | √ | **86.94±3.12** | **85.97±2.19** | **84.28±1.73** | **86.11±2.54** | **87.92±1.69** | **85.12±1.96** |
| Baseline | 40% | 60% | √ | | 74.04±1.21 | 73.72±3.11 | 74.3±2.16 | 73.74±2.94 | 75.61±2.47 | 73.28+2.76 |
| Baseline+SMF-SN | | | √ | | 78.37±2.14 | 77.49±2.76 | 78.06±1.98 | 78.34±2.48 | 79.23±3.16 | 79.35+2.54 |
| Baseline+Gene | | | √ | √ | 78.41±1.43 | 78.16±1.32 | 79.64±2.34 | 78.14±1.79 | 78.02±1.99 | 79.87+1.29 |
| S²MMAM(Ours) | | | √ | √ | **82.35±1.72** | **83.14±1.48** | **83.78±1.77** | 81.87±1.76 | **83.98±1.01** | **83.62+1.52** |
| Baseline | 30% | 70% | √ | | 73.65±2.18 | 72.91±1.03 | 72.11±2.36 | 73.67±2.72 | 73.33±2.31 | 72.51±1.7 |
| Baseline+SMF-SN | | | √ | | 77.29±5.22 | 76.43±4.21 | 77.24±3.21 | 77.17±1.2 | 77.44±5.32 | 78.8±3.69 |
| Baseline+Gene | | | √ | √ | 75.11±2.3 | 78.81±2.92 | 79.35±2.16 | 75.39±3.44 | 75.14±3.26 | 79.08±2.54 |
| S²MMAM(Ours) | | | √ | √ | 81.67±2.67 | 82.31±2.51 | 83.15±1.21 | **82.66±2.07** | 83.27±1.49 | 82.73±1.86 |

features for the classification task, 2) the superiority of multimodal data over single modal data, and 3) the selection of the proportion of labeled images within the training dataset.

We designed three sets of experiments and empirically used data with the proportion of labeled data of 100%, 40%, and 30% as the training dataset. Baseline is used as our base architecture, where Baseline is only constructed by S²MF-CN using CT image data for the classification task. Based on this, we conducted a comparative study by gradually adding SMF-SN, genetic data, and both SMF-SN and genetic data. The experimental results are shown in Table 8.

1) The positive effects of segmentation features for the classification task

As shown in Table 8, better classification results are obtained when the model utilizes the idea of segmentation to facilitate classification. Compared to Baseline, Baseline+SMF-SN improves the AUC values by 6.03%, 3.62%, and 4.11% in 30%, 40%, and 100% labeled datasets, respectively. We also visualize some of our Baseline and Baseline+SMF-SN segmentation results in Fig 9. The results are output in the form of a segmentation graph, which visualizes the ability of the network to localize the lesion area. As can be seen from Fig 9, the model with segmentation task can better localize the lesion area. It can avoid mixing impurities that can easily interfere with the judgment to improve the accuracy of diagnosis.

2) The superiority of multimodal data over single modal data

As shown in Table 8, when we used genetic data, the AUC improved by 3.94%, 2.41%, and 2.81%, respectively, compared with Baseline. This indicates that image data can also extract genotypic features from biological data that can express individual differences and reflect disease characteristics at the micro level. Further, enhances the network information richness and promotes the classification performance.

3) The selection of the proportion of labeled images within the training dataset

As shown in Table 8, when the proportion of labeled data was 30% and 40%, respectively, the difference in the values of the four metrics was small, with a 0.71% difference in AUC and a 0.83% difference in Recall. Compared with the cost of physician labeling, this result indicates that the guidance information contained in 30% labeled training images is sufficient for the

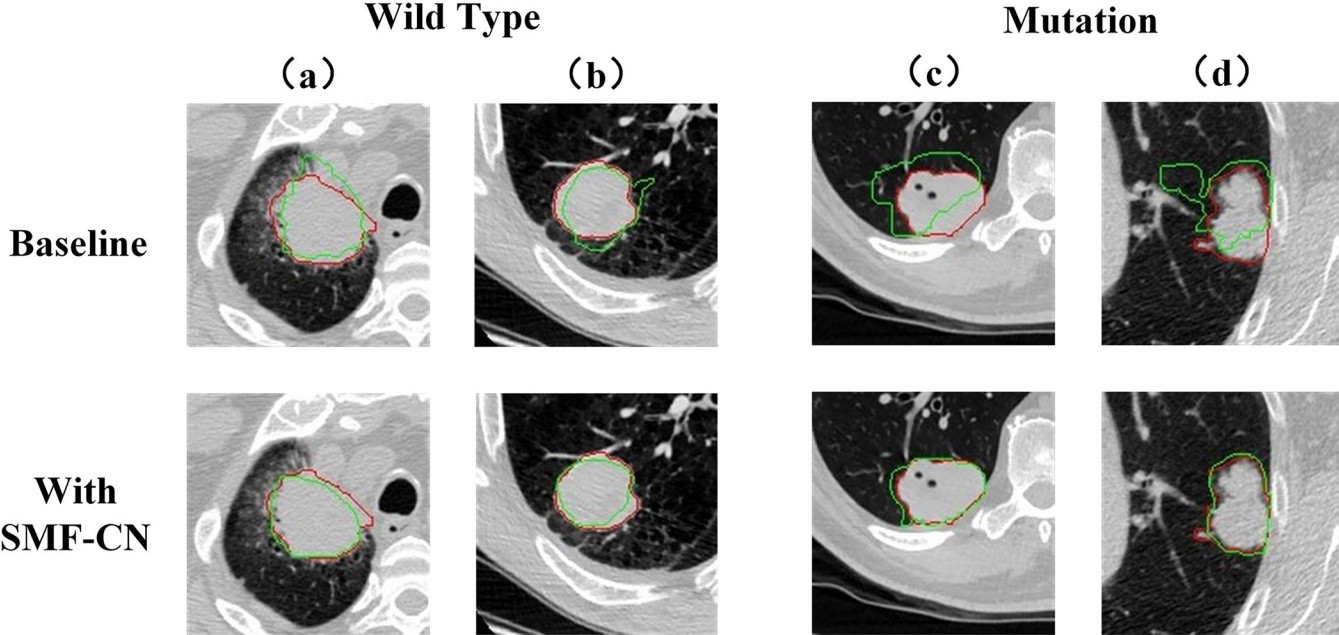

**Fig 9. Comparison of the segmentation results obtained after training on Baseline strategy and Baseline+SMF-SN strategy: Baseline: Only classification task.** Baseline+SMF-SN: classification task and segmentation task. (a) and (b) are the wild type of NSCLC. (c) and (d) are the mutation of NSCLC. The region surrounded by the red line is the ground truth, and the region surrounded by the green line is the segmentation results.

network to learn the key information of the lesion. Therefore, we used 30% labeled images and 70% unlabeled images as the training ratio of the model.

To show the classification performance of our S$^2$MMAM more visually, we also plotted the 3D comparison histograms of AUC and F1 score, as shown in Figs 10 and 11.

In summary, the strategy of sharing segmentation network parameters by the classification network can assist the network to better localize the lesion region. The complementary nature of multimodal data allows the network to learn more abstract features besides addressing the challenge of less information in semi-supervised strategies. Therefore, our S$^2$MMAM is better able to preserve the pathogenic regions, ignore irrelevant information, and improve model sensitivity. This leads to better KRAS mutation prediction results for NSCLC.

## Performance in supervised learning

In order to demonstrate the scalability of our model, our application scenarios will not be limited to semi-supervised learning but will be extended to supervised learning. We compare our S$^2$MMAM with current multimodal classification models that have better results. The competing methods include Multimodal Feature Fusion Diagnostic Model (MFFDM) [39], PLNM [9]. Note that we reproduce the above methods on the same test set for the sake of fairness.

As shown in Table 9, our S$^2$MMAM achieved the best AC, SP, and AUC values. This shows that our model has excellent classification performance even in supervised learning applications. The AUC is 1.6% more than the second place PLNM and 3.75% more than the MFFDM. The fusion method of the MFFDM employs a simple splicing fusion, which we believe is the reason for the poor classification performance. Our S$^2$MMAM employs a multidimensional fusion, which means that it is better able to adaptively fuse complementary information. Our S$^2$MMAM and PLNM are similar in classification performance, but our method achieves better AUC values. We believe that SSL models can achieve the purpose of utilizing limited

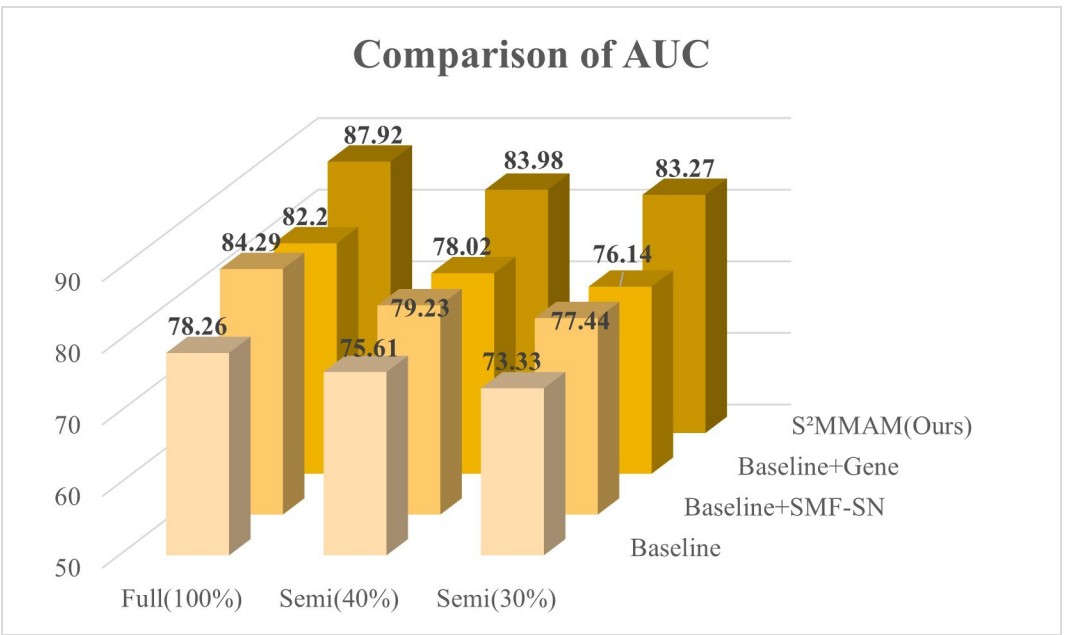

**Fig 10. AUC were achieved on the test set by Baseline, Baseline+SMF-SN, Baseline+Gene and our S²MMAM, when using 30%, 40% and 100% labeled training images.**

information to achieve accurate prediction. When we train with more labeled data, our S²MMAM can have a better ability to extract information and integrate information. In summary, as described, our S²MMAM can be used not only in SSL but also in supervised learning.

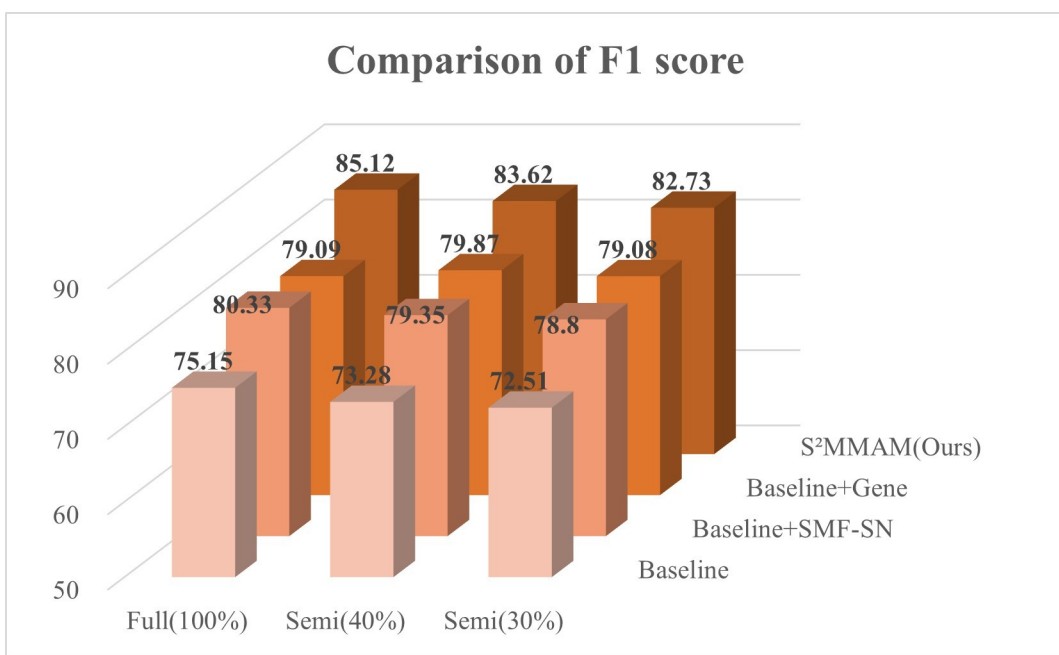

**Fig 11. F1 score were achieved on the test set by Baseline, Baseline+SMF-SN, Baseline+Gene and our S²MMAM, when using 30%, 40% and 100% labeled training images.**

**Table 9. Comparison of the classification performance of S²MMAM and two other supervised medical image classification models.**

| Methods | AC(%) | Recall(%) | Precision(%) | SP(%) | AUC(%) | F1(%) |
|---|---|---|---|---|---|---|
| MFFDM [39] | 84.15±1.45 | 84.22±2.04 | 83.98±2.77 | 84.02±1.97 | 84.17±1.03 | 84.16±1.17 |
| PLNM [9] | 86.34±2.11 | **86.21±2.61** | **85.24±1.69** | 85.73±2.65 | 86.32±1.87 | **85.23±2.31** |
| S²MMAM(Ours) | **86.94±3.12** | 85.97±2.19 | 84.28±1.73 | **86.11±2.54** | **87.92±1.69** | 85.12±1.96 |

It is a non-invasive method to determine whether the KRAS gene is mutated or not, to determine the treatment for patients early, and to improve the survival rate of patients.

## Conclusion

In this paper, we propose an integrating Image and Gene Data with a Semi-Supervised Attention Model for the Prediction of KRAS Gene Mutation Status in Non-Small Cell Lung. The model consists of two components: supervised multilevel fusion segmentation network (SMF-SN) and semi-supervised multimodal fusion classification network (S²MF-CN) fusion. The results on the NSCLC-Radiogenomics dataset demonstrate that S²MMAM can achieve a more accurate prediction of KRAS gene mutation status.

However, our S²MMAM still has some limitations. First, the model tested in this study used a single dataset and was not tested on multiple different datasets. Second, although CT images have been shown to aid in the prediction of KRAS gene mutations. However, in the clinical setting, histopathology images are the gold standard. We will try to combine CT images, histopathology images, and genetic data to further improve the accuracy of KRAS gene mutation status prediction in non-small cell lung cancer.

## Author Contributions

**Conceptualization:** Yuting Xue, Dongxu Zhang, Liye Jia, Ying Qiao, Huajie Yue.

**Data curation:** Dongxu Zhang, Ying Qiao, Huajie Yue.

**Formal analysis:** Yuting Xue, Liye Jia, Wanting Yang, Huajie Yue.

**Funding acquisition:** Juanjuan Zhao, Yan Qiang.

**Investigation:** Liye Jia, Wanting Yang.

**Methodology:** Yuting Xue, Juanjuan Zhao, Long Wang.

**Project administration:** Yan Qiang.

**Resources:** Yan Qiang.

**Software:** Yuting Xue, Dongxu Zhang.

**Supervision:** Long Wang.

**Validation:** Liye Jia, Wanting Yang, Long Wang.

**Writing – original draft:** Yuting Xue.

**Writing – review & editing:** Dongxu Zhang, Wanting Yang, Juanjuan Zhao, Ying Qiao, Huajie Yue.

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
