## [Decision Letter · Decision Letter 0]

22 Aug 2023

PONE-D-23-16921Integrating Image and Gene-Data with a Semi-Supervised Attention Model for Prediction of KRAS Gene Mutation Status in Non-Small Cell Lung CancerPLOS ONE

Dear Dr. Zhao,

Thank you for submitting your manuscript to PLOS ONE. After careful consideration, we feel that it has merit but does not fully meet PLOS ONE’s publication criteria as it currently stands. Therefore, we invite you to submit a revised version of the manuscript that addresses the points raised during the review process.

We look forward to receiving your revised manuscript.

Kind regards,

Jeonghwan Gwak, PhD

Academic Editor

PLOS ONE

“This work was supported by the National Natural Science Foundation of China (Grant No. U21A20469); the National Natural Science Foundation of China (Grant No. 61972274); the Central Government Guides Local Science and Technology Development Fund Project (Grant No. YDZJSX2022C004); the Natural Science Foundation of Shanxi Province (Grant No. 202103021224066); and NHC Key Laboratory of Pneumoconiosis Shanxi China Project, (Grant No.2020-PT320-005), the Non-profit Central Research Institute Fund of Chinese Academy of Medical Science.”

“This work was supported by the National Natural Science Foundation of China (Grant No. U21A20469); the National Natural Science Foundation of China (Grant No. 61972274); the Central Government Guides Local Science and Technology Development Fund Project (Grant No. YDZJSX2022C004); the Natural Science Foundation of Shanxi Province (Grant No. 202103021224066); and NHC Key Laboratory of Pneumoconiosis Shanxi China Project, (Grant No.2020-PT320-005), the Non-profit Central Research Institute Fund of Chinese Academy of Medical Science.”

“This work was supported by the National Natural Science Foundation of China (Grant No. U21A20469); the National Natural Science Foundation of China (Grant No. 61972274); the Central Government Guides Local Science and Technology Development Fund Project (Grant No. YDZJSX2022C004); the Natural Science Foundation of Shanxi Province (Grant No. 202103021224066); and NHC Key Laboratory of Pneumoconiosis Shanxi China Project, (Grant No.2020-PT320-005), the Non-profit Central Research Institute Fund of Chinese Academy of Medical Science.”

Additional Editor Comments:

AE Comments: Thank you for submitting your manuscript. I appreciate the efforts you have put into this research. I have received feedback from the reviewers, and I would like to share their comments and suggestions with you.

1) Clarity and Comprehension: Reviewer 1 points out a lack of clarity in the explanation of your proposed method. The reviewer found it difficult to understand, making it challenging to reproduce the experiments. Specific feedback has been given regarding figures and their captions (Fig. 1, Fig. 2, and Fig. 5), as well as the use of equations (e.g., Equations 8 and 9).

2) Novelty and Originality: Reviewer 2 has raised concerns about the originality of the work. It's essential to clarify the unique contributions of your research compared to existing literature.

3) Related Work: Both reviewers emphasize the need to improve the section on related works. The current version lists existing works without analyzing their limitations. Consider adding a more detailed analysis and perhaps summarizing existing studies in a tabular form to improve readability.

4) Methodology and Experimental Details: Both reviewers have made suggestions to provide more information on the methodology, hyperparameters, network configurations, and a thorough description of the experimental phases.

5) Source Code: Reviewer 2 suggests providing a GitHub link for the source code to enhance repeatability and verification of the study.

6) Grammar and Typos: Both reviewers have found grammatical errors and typos in the manuscript. It is advised to run the manuscript through a grammar checker and proofread it carefully.

7) Additional Feedback: Reviewer 2 has provided an extensive list of recommendations to enhance the quality and clarity of the manuscript. These include improving the introduction, elaborating on tables, addressing overfitting, revisiting results, and ensuring that the references are up-to-date.

In light of the feedback, I recommend revising your manuscript, addressing the concerns raised by the reviewers. This will not only enhance the clarity and quality of your research but also strengthen its contribution to the field.

I hope this feedback is constructive and assists you in enhancing your manuscript. I look forward to receiving your revised submission.

Reviewers' comments:

Reviewer's Responses to Questions

**Comments to the Author**

1. Is the manuscript technically sound, and do the data support the conclusions?

Reviewer #1: Yes

Reviewer #2: Partly

2. Has the statistical analysis been performed appropriately and rigorously? 

Reviewer #1: Yes

Reviewer #2: I Don't Know

3. Have the authors made all data underlying the findings in their manuscript fully available?

Reviewer #1: Yes

Reviewer #2: Yes

4. Is the manuscript presented in an intelligible fashion and written in standard English?

Reviewer #1: Yes

Reviewer #2: No

5. Review Comments to the Author

Reviewer #1: This study proposes a deep learning-based methodology for classifying the oncogenic gene KRAS, which frequently involves non-small cell lung cancer (NSCLC), using CT images and genetic information. Their main contributions are the development of the 'Semi-supervised Multimodal Multiscale Attention' mechanism and the novel 'Attention-guided Feature Aggregation' module. The proposed method appears to be novel and innovative, and the data and analysis seem to fully support their claims. However, due to the lack of clarity in the explanation of the proposed method, it is difficult to understand, making it seem impossible to reproduce the experiments. Therefore, 'minor revisions' are suggested to improve the paper.

In Section 3.1, while it seems that the input of the Supervised Multilevel Fusion Segmentation Network (SMF-SN) is X_L and the output is Y_L, in Fig. 1. (a), it is not clearly introduced what the input and output of SMF-SN are, as both X_L and Y_L are shown. This requires modification.

Overall, the introduction of the proposed system is difficult to comprehend. For example, in Fig. 2, there is only one ASPP block, but the caption suggests the presence of multiple ASPPs. Additionally, in Fig. 1 (b), is the input of the Student Model S_L and the input of the Teacher Model C_U? How are the predictions of the Student and Teacher integrated?

In Fig. 5, an explanation is needed for 'Genes Selection'.

The paper requires revisions for grammatical errors.

Equations must be used in the right way (e.g. Equations 8 and 9 should have the first two letters of Recall and Precision in italics.)

Reviewer #2: The experimental study is interesting information in this paper. However, the main weakness of the paper lies in its lack of originality and novelty. The following suggestions may be considered to enhance the quality and clarity of the manuscript

1- The motivation is not clear. Why did this work? Is any problem does it address that the previous methods cannot?

2- The introduction section could be improved by clarifying the similarities and differences between the related work and the proposed method are not clearly described. It is recommended to add a separate subsection and clear description in this regard.

3- Related work: The paper only lists existing works in the research community without any analysis of existing work's limitations. Therefore, I suggest that the authors mention more summary and limitation analysis so that readers can easily appreciate the contributions made by this paper.

4- In the related works, existing studies can also be summarized in a tabular form to improve readability

5- Elaborate all tables briefly.

6- How to deal with overfitting in your model?

7- Results and illustrations need to be revisited.

8- Background information of this work can be provided more systematically and comprehensively, i.e. logic of this paper should be further enhanced.

9- - Hyperparameters of the model:

- The initialization method is not mentioned.

10- Similarly, the network configurations can be summarized in a table e.g. input size, # of layers, learning rate, optimizers etc.

11- Furthermore, the study's application is not explained in an intelligible manner. You should include an experimentation section to provide readers with a thorough description of all the experimental phases in a straightforward and accessible manner.

12- The theoretical and practical sections of the study are not adequately convincing, and the writing style is absolutely insufficient to highlight the subjective contribution to your research when compared to past research findings.

13- Another important aspect of scientific research is the capacity to repeat the experiment or study in a different setting and reuse or adapt the findings. This is an important point, and you could elaborate on it further in the discussion area to give additional scientific value to this critical study.

14- Please include a link in the research article that allows the complete applied side of this study to be downloaded for verification, validation, and inspection, as well as so that it may be used as a scientific reference.

The code source of this work must be added as a comment to the paper and must be uploaded as a GitHub link to be visible and referenceable.

15- In addition to these specific recommendations, the authors should also run the manuscript through a grammar checker like Grammarly to address any language or grammatical errors. Finally, the authors should ensure that all references cited in the manuscript are up-to-date and relevant to the research topic.

16- Typos/Grammatical Errors:

Subsection Segmentation facilitates classification

Deep Convolutional Nneural Networks  N should be removed from neural

Section Conclusion:

Mutation Status in Non-Small Cell Lung The model  period (.) is missing

network (S2 MF-CN). fusion.  the extra period (.) should be removed

6. PLOS authors have the option to publish the peer review history of their article (what does this mean?). If published, this will include your full peer review and any attached files.

Reviewer #1: No

Reviewer #2: No

---

## [Author Response · Author response to Decision Letter 0]

13 Oct 2023

Responses to Reviewers' 

Dear Editors and Reviewers, 

Thank you for your letter and comments on our manuscript entitled "Integrating Image and Gene-Data with a Semi-Supervised Attention Model for Prediction of KRAS Gene Mutation Status in Non-Small Cell Lung Cancer (ID: PONE-D-23-16921)". We sincerely thank all reviewers for their time and effort. According to the constructive comments of the editors and reviewers on improving the quality of the revised version of this paper, we have revised the whole manuscript carefully and tried to avoid any grammar or syntax errors. In addition, we have asked several colleagues who are skilled in English papers to help us thoroughly check the organization and language of the paper. We hope for accepting our further improved submitted paper for possible publication in the PLOS ONE distinguished journal.We have revised the manuscript point by point. We apologize for not using "the Tracked Changes function in Word." The reason is that we revised all grammar or syntax errors and made a lot of changes, which might have interfered with the editors and reviewers reviewing the paper. We have highlighted the important changes in red. We hope you will be satisfied with our revised manuscript. Responses to comments, as well as details of revisions, are given below.

Sincerely yours, 

Juanjuan Zhao (on behalf of all the co-authors)

Reviewer #1:

Comment1:

In Section 3.1, while it seems that the input of the Supervised Multilevel Fusion Segmentation Network (SMF-SN) is X_L and the output is Y_L, in Fig. 1. (a), it is not clearly introduced what the input and output of SMF-SN are, as both X_L and Y_L are shown. This requires modification.

Response1:

We sincerely thank the reviewers for asking rigorous questions.

We feel very sorry that we lacked some explanations here. The input of SMF-SN is CT images, which is , and pixel-level mask images annotated by physicians, which is . The output of SMF-SN is the segmented lesion map. Since segmentation performance is not a concern in this study, it is not highlighted in the figure. We have added explanations for input and output in the caption of Fig. 1(a) in new manuscript to improve readability.

Comment2:

Overall, the introduction of the proposed system is difficult to comprehend. For example, in Fig. 2, there is only one ASPP block, but the caption suggests the presence of multiple ASPPs. Additionally, in Fig. 1 (b), is the input of the Student Model S_L and the input of the Teacher Model C_U? How are the predictions of the Student and Teacher integrated?

Response2:

Sorry we sincerely appreciate these insightful questions and apologize for our lack of rigor.

(1)We apologize for our carelessness.We have changed ‘ASPPs’ to ‘ASPP’ in new manuscript. 

(2)and (3)

Fig.1 The Mean Teacher method. The figure depicts a training batch with a single labeled example. Both the student and the teacher model evaluate the input applying noise (η, η’) within their computation. The softmax output of the student model is compared with the one-hot label using classification cost and with the teacher output using consistency cost. After the weights of the student model have been updated with gradient descent, the teacher model weights are updated as an exponential moving average of the student weights. Both model outputs can be used for prediction.

We very much apologize for not articulating this clearly. Fig.1 comes from the paper Mean Teacher Model [1]. According to the description in the paper, the inputs to the Mean Teacher Model are labeled and unlabeled data . The principle of the model is that first, the labeled data and unlabeled data are trained in the Student model. The labeled dataset will produce a classification loss L1. The unlabeled dataset will produce a prediction p1. Then, the unlabeled dataset is put into the Teacher model for training and produces a prediction p2. At this time, a distribution consistency loss L2 is computed, which is the difference between p1 and p2. At this point, the total loss L of the network is calculated as the sum of the L1 and L2. According to the L, the network parameters θ of the Student model are updated. Based on the EMA algorithm, θ will update the parameters θ’ of the Teacher model. After training, the model prediction performance is improved, and it can perform accurate predictions on unlabeled data.

Based on your suggestion, We redraw Fig. 1 to clearly show the inputs to the model. means labled dataset for segmentation. means labled dataset for classification. means unlabled dataset for classification. is the input of SMF-SN but not for classification network.

Comment3:

In Fig. 5, an explanation is needed for 'Genes Selection'.

Response3:

We sincerely thank the reviewers for the detailed comments.

We explained a detailed description of 'Genes Selection' in Section Data preprocessing. However, in Section Data preprocessing, our caption was set to 'Gene Data' in the previous version of the manuscript, causing ambiguity. We feel very sorry for this and have changed 'Gene Data' to 'Genes Selection' in the new version of the manuscript to prevent ambiguity.

Comment4:

The paper requires revisions for grammatical errors.

Response4:

We sincerely thank the reviewers for the detailed comments.

According to your suggestion, We have checked our manuscript carefully and corrected the grammatical, styling, and typos found in our new manuscript. Moreover, we have asked several colleagues who are skilled in English papers to help us thoroughly check the organization and language of the paper. 

Comment5:

Equations must be used in the right way (e.g. Equations 8 and 9 should have the first two letters of Recall and Precision in italics.)

Response5:

We sincerely thank the reviewers for this insightful question.

We have checked all the Equations carefully for formatting issues and made corrections.

Reviewer #2:

Comment1:

The motivation is not clear. Why did this work? Is any problem does it address that the previous methods cannot?

Response1:

We sincerely thank the reviewers for asking rigorous questions.

We have supplemented a detailed description of the motivation for the research in the new manuscript (Section Introduction, P1, L3-L4). The emergence of targeted therapy has substantially increased the survival rate of NSCLC patients. Mutations of essential pathogenic genes should be identified before targeted therapy. KRAS is a gene type with a high probability of mutation. It is necessary for diagnosing whether a patient has a KRAS gene mutation.

We listed the limitations of the previous methodology in the fourth paragraph of Section Introduction. The main solution of this study is the three problems listed. 1)Majority of deep learning methods that study classification tasks focus only on methods for classification. However, these studies did not use the segmentation features generated by the segmentation task to facilitate the classification task to improve the performance and effectiveness of the classification task. 2) Most of the studied fusion methods used simple fusion means of direct concatenation. But, they ignore the correlation and difference between medical images and genetic data. It not only leads to ineffective mining of useful semantic features between multi-scale image features and gene features, but also fails to make full use of the complementarity of multimodal information. 3) Many studies used models that overemphasized the deep features of lesion abstraction. Nonetheless, they did not pay sufficient attention to the importance of detailed shallow features in prediction results. This leads to limitations in improving accuracy.

Comment2:

The introduction section could be improved by clarifying the similarities and differences between the related work and the proposed method are not clearly described. It is recommended to add a separate subsection and clear description in this regard.

Response2:

We feel great thanks for your professional review work on our paper.

We have added a new subsection according to the your suggestion in sixth paragraph of Introduction section to further compare the similarities and differences between previous work and ours. 

Comment3:

Related work: The paper only lists existing works in the research community without any analysis of existing work's limitations. Therefore, I suggest that the authors mention more summary and limitation analysis so that readers can easily appreciate the contributions made by this paper.

Response3:

Thank you very much for the professional review work you have done on our papers.

Following your third and fourth comments, we have summarized and compared past work in a tabular format. The reason for not tabulating the comparison in the section Multiscale features and attention learning is that we consider both approaches classic and valid. We are concerned that the papers listed above all focus on only one aspect. Our contribution is to combine both methods to obtain better performance in extracting lesion information.

Comment4:

In the related works, existing studies can also be summarized in a tabular form to improve readability.

Response4:

Thank you very much for the professional review work you have done on our papers.

Following your third and fourth comments, we have summarized and compared past work in a tabular format to improve readability.

Comment5:

Elaborate all tables briefly.

Response5:

We feel great thanks for your professional review work on our paper.

Based on your suggestions, we have further elaborated all the tables for a better understanding of the readers.

Comment6:

How to deal with overfitting in your model?

Response6:

We feel great thanks for your professional review work on our paper. 

We mainly used a cross-validation approach to prevent overfitting problems. We set the cross-validation to 5-fold, 10-fold and 15-fold. Table 1 records the AUC values of the test dataset at different parameter settings and with different scales of labeled data. The results show that the test dataset has the highest accuracy when the cross-validation is equal to 10-fold.

Table 1. AUC values of the test dataset under different parameter settings for different scales of labeled data.

Setting 5-fold 10-fold 15-fold

100% Labeled 84.16 87.92 88.76

40% Labeled 78.64 83.98 82.44

30% Labeled 70.02 83.27 80.69

Comment7:

Results and illustrations need to be revisited.

Response7:

Thank you very much for your professional review of our paper. 

We feel sorry for our lack of rigor. We have re-examined the results and illustrations. We found that some of the illustrations' descriptions did not match the illustrations' content, as in Fig 8. We have corrected the incorrect parts and confirmed that the results and illustrations are correct. We will pay more attention to the uploading requirements to ensure readers see the correct results and illustrations.

Comment8:

Background information of this work can be provided more systematically and comprehensively, i.e. logic of this paper should be further enhanced.

Response8:

We feel great thanks for your professional review work on our paper. 

Based on your suggestions, we have reorganized the logic of the article and partially rewritten it to make it present the objectives of the study more clearly.

Comment9:

Hyper-parameters of the model:The initialization method is not mentioned.

Response9:

We feel great thanks for your professional review work on our paper. 

The initialization of hyper-parameters is mentioned in the Implementation details of Section Implementation details. The following is the initialization information for the hyper-parameters: All models in the experiments are trained using 10-fold cross-validation, with the number of epochs per fold set to 20. Adams was used as our optimizer. The initial learning rate is set to 0.001 empirically, and the batch size is set to 16.

Comment10:

Similarly, the network configurations can be summarized in a table e.g. input size, # of layers, learning rate, optimizers etc.

Response10:

We sincerely thank the reviewers for asking rigorous questions. 

Based on your suggestion, we have redrawn Fig. 2. In Fig. 2, we have added the basic parameters of the network, such as input size. Since SMF-SN and S2MF-CN have the same structure, there is no separate structural diagram for S2MF-CN.

According to your suggestion, we design the hyper-parameter contents to be summarized in the form of Table 4 to improve readability.

Comment11:

Furthermore, the study's application is not explained in an intelligible manner. You should include an experimentation section to provide readers with a thorough description of all the experimental phases in a straightforward and accessible manner.

Response11:

Thank you very much for your advice. 

We have added the visualization of the results of the experimental procedure in the Section Superiority of the model. The visualization shows the focus and significance of our study in an intuitive way. We believe that readers can understand the advantages of our method in this way.

Comment12:

The theoretical and practical sections of the study are not adequately convincing, and the writing style is absolutely insufficient to highlight the subjective contribution to your research when compared to past research findings.

Response12:

We feel great thanks for your professional review work on our paper.

In the theoretical part, we have comprehensively revised the logic of the paper with the above comments to highlight the superiority of our method.

In the experimental part, we have further analyzed the experimental results according to your suggestions. Detailed descriptions are provided in Sections Ablation studies and Sections Comparison experiment, to better demonstrate the good classification performance of our model. In the Discussion section, we have reorganized the logic to highlight the advantages of the model in a more logical form. In addition, according to #comment13, we have also supplemented extended experiments in the Discussion section to demonstrate the expandability and reusability of the experiments. 

Through the above methods, we hope to make the paper more convincing.

Comment13:

Another important aspect of scientific research is the capacity to repeat the experiment or study in a different setting and reuse or adapt the findings. This is an important point, and you could elaborate on it further in the discussion area to give additional scientific value to this critical study.

Response13:

We feel great thanks for your professional review work on our paper.

Following your suggestion, we have added 'Performance in Supervised Learning' in Section Performance in Supervised Learning to demonstrate the extensibility of our model. The experimental proof demonstrates that our model is not only applicable in semi-supervised learning but can also be used in supervised learning. This shows that our research can be realized in various application scenarios.

Comment14:

 Please include a link in the research article that allows the complete applied side of this study to be downloaded for verification, validation, and inspection, as well as so that it may be used as a scientific reference.The code source of this work must be added as a comment to the paper and must be uploaded as a GitHub link to be visible and referenceable.

Response14:

We feel great thanks for your professional review work on our paper.

We've included a link to the code in the Data availability.

The specific modifications are as follows:

The data are available from the website

https://wiki.cancerimagingarchive.net/display/Public/NSCLC+Radiogenomics. The code for S2MMAM is available on a GitHub repository at https://github.com/xyttttboom/SSMMAM.

Comment15:

In addition to these specific recommendations, the authors should also run the manuscript through a grammar checker like Grammarly to address any language or grammatical errors. Finally, the authors should ensure that all references cited in the manuscript are up-to-date and relevant to the research topic.

Response15:

We sincerely appreciate these insightful questions and apologize for our lack of rigor. 

According to your suggestion, we have used Grammarly to address all language and grammatical errors.Moreover, we have asked several colleagues who are skilled in English papers to help us thoroughly check the organization and language of the paper.

We rechecked the references, deleting papers with little relevance to the topic and adding new papers with vital relevance. In Section Comparison experiment, we also rechecked the literature and compared it with recently published papers with better results.

e.g.

34.Cai M, Zhao L, Zhang Y, Wu W, Jia L, Zhao J, Yang Q, Qiang Y. A progressive phased attention model fused histopathology image features and gene features for lung cancer staging prediction. Int J Comput Assist Radiol Surg. 2023 Oct;18(10):1857-1865. 

39.Zeng LL, Gao K, Hu D, Feng Z, Hou C, Rong P, Wang W. SS-TBN: A Semi-Supervised Tri-Branch Network for COVID-19 Screening and Lesion Segmentation. IEEE Trans Pattern Anal Mach Intell. 2023 Aug;45(8):10427-10442.

40.Chen X, Bai Y, Wang P, Luo J. Data augmentation based semi-supervised method to improve COVID-19 CT classification. Math Biosci Eng. 2023 Feb 6;20(4):6838-6852. doi: 10.3934/mbe.2023294. 

41.Tu Y, Lin S, Qiao J, et al. Alzheimer’s disease diagnosis via multimodal feature fusion. Computers in Biology and Medicine, 2022, 148: 105901.

Comment16:

Typos/Grammatical Errors:

Subsection Segmentation facilitates classification

Deep Convolutional Nneural Networks  N should be removed from neural

Section Conclusion:

Mutation Status in Non-Small Cell Lung The model  period (.) is missing

network (S2MF-CN). fusion.  the extra period (.) should be removed.

Response16:

We sincerely appreciate these insightful questions and apologize for our lack of rigor. 

Based on your comments, we have made the corrections to revise the typos and grammatical errors throughout the paper.

References:

[1]arvainen A, Valpola H. Mean teachers are better role models: Weight-averaged consistency targets improve semi-supervised deep learning results. Adv Neural Inf Process Syst. 2017;30. doi:10.5555/3294771.3294885.

---

## [Decision Letter · Decision Letter 1]

3 Jan 2024

Integrating Image and Gene-Data with a Semi-Supervised Attention Model for Prediction of KRAS Gene Mutation Status in Non-Small Cell Lung Cancer

PONE-D-23-16921R1

Dear Dr. Zhao,

We’re pleased to inform you that your manuscript has been judged scientifically suitable for publication and will be formally accepted for publication once it meets all outstanding technical requirements.

Kind regards,

Jeonghwan Gwak, PhD

Academic Editor

PLOS ONE

Additional Editor Comments (optional):

AE: After careful consideration and based on the insightful feedback from our reviewers, I am delighted to announce that your paper is now deemed publishable.

Reviewers' comments:

Reviewer's Responses to Questions

**Comments to the Author**

1. If the authors have adequately addressed your comments raised in a previous round of review and you feel that this manuscript is now acceptable for publication, you may indicate that here to bypass the “Comments to the Author” section, enter your conflict of interest statement in the “Confidential to Editor” section, and submit your "Accept" recommendation.

Reviewer #2: All comments have been addressed

2. Is the manuscript technically sound, and do the data support the conclusions?

Reviewer #2: Yes

3. Has the statistical analysis been performed appropriately and rigorously? 

Reviewer #2: Yes

4. Have the authors made all data underlying the findings in their manuscript fully available?

Reviewer #2: Yes

5. Is the manuscript presented in an intelligible fashion and written in standard English?

Reviewer #2: Yes

6. Review Comments to the Author

Reviewer #2: (No Response)

7. PLOS authors have the option to publish the peer review history of their article (what does this mean?). If published, this will include your full peer review and any attached files.

Reviewer #2: **Yes: **Zahid Ullah

---

## [Editor Report · Acceptance letter]

1 Mar 2024

PONE-D-23-16921R1 

PLOS ONE

Dear Dr. Zhao, 

I'm pleased to inform you that your manuscript has been deemed suitable for publication in PLOS ONE. Congratulations! Your manuscript is now being handed over to our production team.

Kind regards, 

on behalf of

Dr. Jeonghwan Gwak 

Academic Editor

PLOS ONE